STEM CELLS AND REGENERATION

# Crip2 preserves hematopoietic stem and progenitor cell production through inhibition of Notch signals

Angelika G. Aleman[1], Bianca Ulloa[2,3], Rigolin Nayak[4], Caitlin Ford[5], Kathryn S. Potts[2,3], Carmen de Sena-Tomás[4,6], Camila Vicioso[4], Uday Rangaswamy[7], Harold K. Elias[8], Michael G. Kharas[8], Remo Sanges[7,9], Teresa Bowman[2,3,10] and Kimara L. Targoff[4,11,*]

## ABSTRACT

Hematopoietic stem and progenitor cells (HSPCs) have multilineage potential and sustain long-term self-renewal. Deriving patient-specific HSPCs has immense therapeutic potential to overcome the shortage of compatible donors for transplantation. In zebrafish, hemogenic endothelium (HE) is a specialized collection of dorsal aortic endothelial cells (ECs) that give rise to HSPCs. Our data reveal that Cysteine rich intestinal protein 2 (Crip2) has a previously unrecognized function in establishing the proper EC environment for HSPC specification. To investigate the requirement of Crip2, we generated loss-of-function alleles in *crip2* and *crip3*, a gene family member with cardiovascular expression. *crip2*⁻/⁻;*crip3*⁻/⁻ (*crip*^DM) embryos exhibit decreased HSPC emergence with impaired lineage derivative production. Single cell RNA-sequencing of *kdrl*:mCherry⁺ ECs reveals upregulation of vascular development signature and failure to repress Notch signals during the vital transition of HE specification to HSPC emergence. Moreover, our data underscore that inhibition of Notch promotes HSPC generation in *crip*^DM embryos and Crip genes operate through NF-κB to limit Notch. Identification of Crip2 as a novel regulator of Notch repression in HE will enhance our understanding of cues necessary to improve human HSPC production *in vitro*.

KEY WORDS: Crip2, HSPC, Notch, Hemogenic endothelium, Zebrafish

[1]Department of Physiology & Cellular Biophysics, College of Physicians & Surgeons, Columbia University, New York, NY 10032, USA. [2]Department of Developmental and Molecular Biology, Albert Einstein College of Medicine, Bronx, NY 10461, USA. [3]Gottesman Institute of Stem Cell Biology and Regenerative Medicine, Albert Einstein College of Medicine, Bronx, NY 10461, USA. [4]Division of Cardiology, Department of Pediatrics, College of Physicians & Surgeons, Columbia University, New York, NY 10032, USA. [5]Department of Genetics & Development, College of Physicians & Surgeons, Columbia University, New York, NY 10032, USA. [6]Department of Genetics, Microbiology and Statistics, University of Barcelona, 08028 Barcelona, Spain. [7]Functional and Structural Genomics, Scuola Internazionale Superiore di Studi Avanzati (SISSA), Via Bonomea 265, 34136 Trieste, Italy. [8]Molecular Pharmacology Program, Sloan Kettering Institute, Memorial Sloan Kettering Cancer Center, New York, NY 10065, USA. [9]Central RNA Laboratory, Istituto Italiano di Tecnologia (IIT), Via Enrico Melen 83, 16152 Genova, Italy. [10]Department of Medicine (Oncology), Albert Einstein College of Medicine and Montefiore Medical Center, Bronx, NY 10461, USA. [11]Columbia Stem Cell Initiative, Columbia University, New York, NY 10032, USA.

*Author for correspondence (kl284@columbia.edu)

 K.L.T., 0000-0002-6066-6002

## INTRODUCTION

Hematopoietic stem and progenitor cells (HSPCs) establish the blood system and serve as the source of many cell types given their ability for self-renewal and differentiation into mature lineages. Significant advances have been made in identifying transcriptional networks responsible for HSPC ontogeny. Moreover, stem cell-based regenerative and transplantation medicine is a therapeutic necessity for both malignant and non-malignant hematopoietic diseases (Castagnoli et al., 2019; Dazzi et al., 2007; Delaney et al., 2016; Gross et al., 2010; Rao et al., 2022). Yet, scalable production of long-lived multi-lineage hematopoietic cells with engraftment potential remains challenging (Slukvin and Uenishi, 2019; Wang and Sugimura, 2023). Despite the burden of human hematopoietic disorders, the ability to drive patient-specific HSPC production from mesodermal progenitors *in vitro* continues to be devastatingly inefficient (Ditadi et al., 2015; Doulatov et al., 2013; Sugimura et al., 2017). Recent advances have demonstrated an ability to deliver vector- and stroma-free transplantable HSPCs; however, the requirement of markers for endothelial and hematopoietic enrichment still limits translation to clinical practice (Piau et al., 2023). Therefore, dissecting the molecular cues that are essential in promoting and sustaining the self-renewing HSPC lineage is crucial to improve this therapeutic potential.

HSPCs are first produced along the ventral wall of the dorsal aorta (VDA) in a specialized hemogenic endothelium (HE) in all vertebrates (Bertrand et al., 2010; Dzierzak and Speck, 2008; Kissa and Herbomel, 2010). In the zebrafish aorta, HE is labeled by *gata2b* expression and subsequent upregulation of *runx1* expression by 24 h post fertilization (hpf) (Butko et al., 2015). HSPCs arise from the floor of the dorsal aorta through an 'endothelial-to-hematopoietic' transition (EHT) (Bertrand et al., 2010; Kissa and Herbomel, 2010; Lam et al., 2010). This process of 'budding' occurs between 30 and 36 hpf, with their emergence into circulation between 40 and 52 hpf (Kissa and Herbomel, 2010). The HSPCs travel to a vascular plexus called the caudal hematopoietic tissue (CHT), a counterpart to the mammalian fetal liver, where they settle and expand for 3-4 days before migrating to the adult hematopoietic niche in the kidney marrow and thymus (Hagedorn et al., 2023; Li et al., 2018; Murayama et al., 2006; Tamplin et al., 2015). Employing a Brainbow-based multicolor Zebrabow system, *in vivo* analysis reveals that 20-30 HSPC clones originating from the HE generate the adult blood system (Henninger et al., 2017). Despite this tightly orchestrated process, the regulators of the HE specification are poorly understood (Slukvin and Uenishi, 2019). Further investigation of the mechanisms guiding the transition from arterial endothelial fate to HE identity will augment our ability to produce renewable and engraftable HSPCs *in vitro* for clinical applications.

We aimed to investigate these mechanisms, and identified *cysteine rich intestinal protein 2* (*crip2*) as a previously

unrecognized gene essential for embryonic hematopoiesis. Crip2 is a LIM-domain-only protein that has reported roles in smooth muscle cell differentiation and migration (Chen et al., 2013; Kihara et al., 2011), tumor suppression (Cheung et al., 2011) and apoptosis (Lo et al., 2012). Murine data has illustrated *Crip2* expression in cardiomyocytes (CMs) and endothelial cells (ECs) (Wei et al., 2011). In zebrafish embryos, *crip2* is maternally expressed and transcripts become localized to the heart tube at 24 hpf and to the atrioventricular canal by 48 hpf (Kim et al., 2014). Furthermore, single cell RNA-sequencing (scRNA-seq) data from adult zebrafish delineate *crip2* expression at the injury border zone in the regenerating heart (Honkoop et al., 2019). More recently, from a lineage-tracing approach to track HSPCs *in vivo* using the regulatory elements of the zebrafish *draculin* (*drl*) gene, *crip2* is identified in the pre-HE (Ulloa et al., 2021), a cell population responsible for initiating an EHT transition (Zovein et al., 2008). Despite these reports hinting at *Crip2* function in cellular processes associated with vascular EC function and HE production, a requirement for developmental hematopoiesis has yet to be uncovered.

Although several transcription factors and signaling pathways exhibit notable roles in enabling the shift from arterial program to hematopoietic identity, Notch activity has emerged as a major director of HSPC emergence (Bigas et al., 2013; Gama-Norton et al., 2015; Lomeli and Castillo-Castellanos, 2020). Since Notch signaling molecules and targets have been shown to be expressed in arterial endothelium, HE and HSPCs (Fadlullah et al., 2022; Zhu et al., 2020), it has been arduous to uncouple the requirements for this crucial cell fate determination pathway in distinguishing arterial identity from hematopoietic commitment (Thambyrajah and Bigas, 2022). However, evidence points to a temporally-controlled downregulation of Notch following specification of HE to boost HSPC departure (Gama-Norton et al., 2015; Lizama et al., 2015; Zhang et al., 2017, 2015). Yet, little is known regarding the upstream regulators responsible for delicately titrating Notch signals. How does Notch transition from its key role in establishing the hemogenic capacity to its repressive function in emerging HSPCs? Our studies underline the innovative finding that *crip2* operates as a negative regulator of Notch specifically as HSPCs arise from the HE. We extend these investigations to include the previously reported physical interaction of Crip2 with NF-κB to propose a mechanism by which Crip genes restrict Notch signals (Cheung et al., 2011). Extrapolation of these insights to *ex vivo* HSPC generation techniques offers strategies for modulating Notch in a spatiotemporal fashion and for improving life-saving applications for hematopoietic disorders more broadly.

In the present study, we reveal a previously unknown role for Crip2 in repression of arterial fate to promote HE specification and foster HSPC emergence during the initial phase of definitive hematopoiesis. We designed a novel loss-of-function model for Crip genes, demonstrating the vital role of *crip2* in embryonic hematopoiesis. Our studies highlight impaired production of HSPCs in the zebrafish dorsal aorta specifically during the first 3 days of blood development. Augmentation of the arterial signature and enhanced expression of Notch pathway components reveal the requirement for Crip2 in appropriate inhibition of these genetic programs. Through repression of elevated Notch signals during the critical window of HSPC emergence, we observe a rescue of hematopoietic efficiency in the Crip loss-of-function model. Moreover, inhibition of NF-κB similarly abrogates the decrement in HSPCs in the context of Notch suppression. Taken together, our findings uncover Crip2 as a new player contributing to HSPC production, its expression being finely tuned to repress Notch signals at the crucial juncture of HSPC emergence. These results from the

zebrafish embryo can inform tactics to heighten the efficiency of HSPC *in vitro* differentiation for therapies of hematopoietic disease.

## RESULTS

### Crip loss-of-function embryos serve as a new model to study early hematopoietic development

Through mining an unpublished bulk RNA-seq dataset performed at 26 hpf, we screened for previously unappreciated, critical regulators of hematopoiesis and identified *crip2* expression in the somites, dorsal aorta and linear heart tube (Fig. 1A-C; Fig. S1A,C). The Crip family constitutes a subclass of LIM-domain-only proteins that includes Crip1 (Birkenmeier and Gordon, 1986), Crip2 (Yu et al., 2002) and Crip3 (Kirchner et al., 2001). These LIM-domain-only proteins are evolutionarily conserved and resemble another family called cysteine- and glycine-rich proteins (Csrp) that are expressed in differentiated vascular smooth muscle cells (Henderson et al., 2002; Jain et al., 1998). Extensive studies have highlighted the importance of HE in the VDA in producing HSPCs (Bertrand et al., 2010; Dzierzak and Speck, 2008; Kissa and Herbomel, 2010). Moreover, recent studies also examine the somitic origins of stromal cell progenitors required for HSPC production in CHT (Murayama et al., 2015, 2023). Thus, given that we first detected *crip2* transcripts in the caudal somites and the dorsal aorta with persistence in the vasculature to 36 hpf (Fig. 1A-C), we conclude that this spatiotemporal configuration suggests potential functions in early hematopoiesis.

Although *crip2* is the only Crip gene family member noted in the caudal somites and dorsal aorta, we observed cardiac expression of both *crip2* and *crip3* in the developing heart (Fig. S1A-E). *crip2* is expressed throughout the myocardium at the linear heart tube stage (26 hpf) and in the atrioventricular canal (AVC) and outflow tract (OFT) following chamber emergence (52 hpf) (Fig. S1C,D). While *crip3* transcripts were identified in the cardiac chambers at 52 hpf (Fig. S1E), *crip1* was only found in the pronephros and faintly in the pharyngeal arches (Fig. S1I,J). To avoid potential overlapping roles with members of the Crip gene family detected in the cardiovascular structures, we targeted both *crip2* and *crip3* with CRISPR multiplex RNA-guided Cas9 nuclease genome editing in zebrafish (Blackburn et al., 2013; Cong et al., 2013; Gagnon et al., 2014; Hwang et al., 2013; Jao et al., 2013). Successful recovery of a nonsense allele for *crip2^fcu2* was identified with a 5 bp deletion in exon 2, yielding a truncated 45 aa protein (Fig. 1G). Similarly, the *crip3^fcu3* allele was generated through an 11 bp deletion in exon 2, yielding a truncated, 63 aa protein (Fig. 1H). Both *crip2^fcu2* and *crip3^fcu3* are predicted to cause truncations in the first LIM domain of each protein, abrogating functional interactions. We benefited from these novel genetic reagents to generate a double mutant, *crip2^{-/-};crip3^{-/-}* (hereafter referred to as *crip^DM*), zebrafish line. Our data illuminate significant downregulation of both targeted Crip genes by *in situ* hybridization (ISH) (Fig. 1D-F; Fig. S1F-H) and qPCR (Fig. 1K,L).

Although *crip^DM* embryos appeared to be normal and were viable (Fig. 1I,J), we scrutinized for potential cardiac development phenotypes that might obfuscate our examination of hematopoiesis in these animals. Surprisingly, we ascertained normal ventricular and atrial morphogenesis and chamber identity maintenance (Fig. S2A-C), cardiomyocyte cell numbers (Fig. S2D-F), and OFT and AVC formation (Fig. S2G-L). These data underscore that *crip2* and *crip3* expression in the heart does not play a noteworthy role in zebrafish cardiac morphogenesis. Altogether, we conclude that *crip^DM* embryos serve as an excellent model to probe the function of *crip2* in early hematopoietic development.

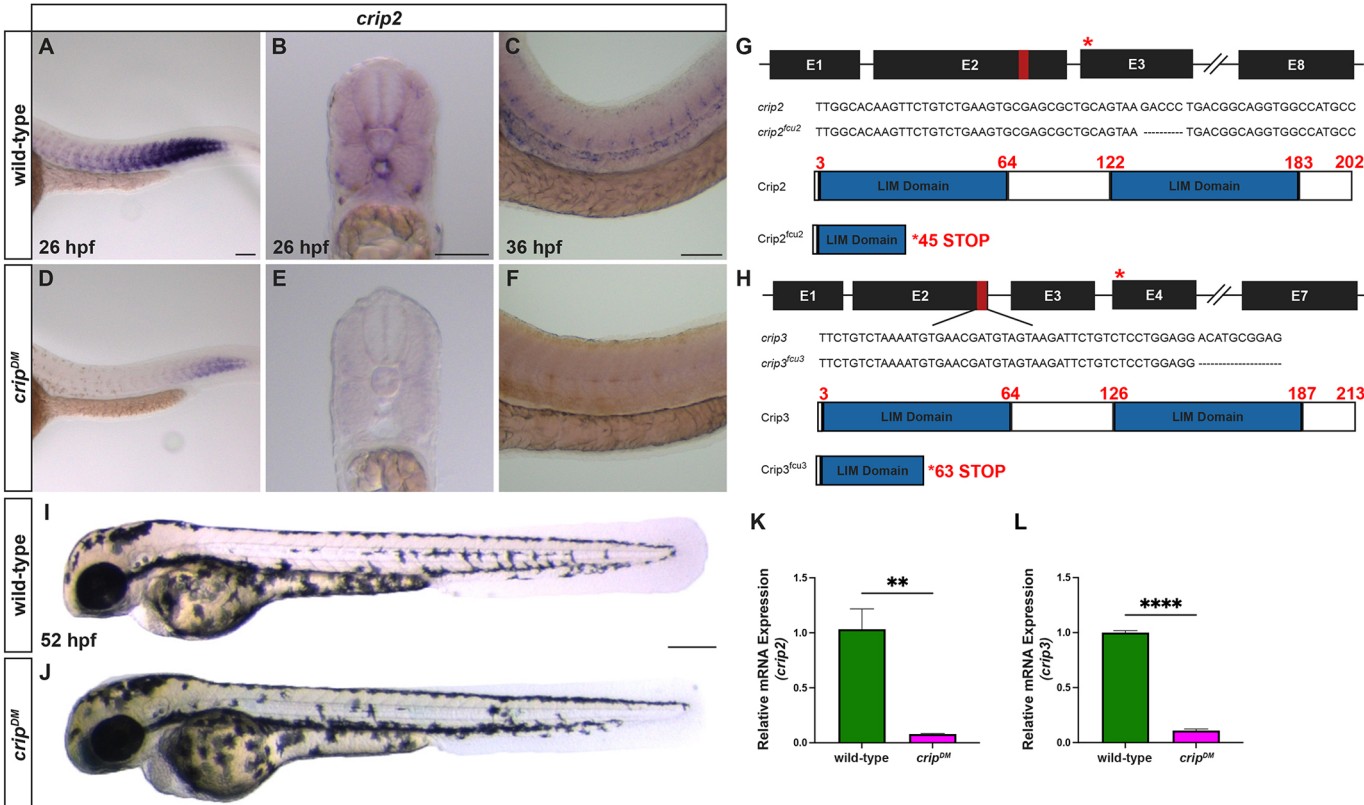

**Fig. 1. *crip2* is expressed in the endothelium of the dorsal aorta.** (A-F) *In situ* hybridization for *crip2* in wild-type (*n*=35) (A), (*n*=15) (B), (*n*=20) (C) and in *crip*$^{DM}$ (*n*=35) (D), (*n*=15) (E), (*n*=20) (F) embryos at 26 hpf (A,B,D,E) and 36 hpf (C,F). Lateral views, anterior to the left (A,C,D,F). Transverse views, dorsal to the top (B,E). Scale bars: 100 µm. (G) Schematic depicts the zebrafish *crip2* gene in black and the wild-type and predicted *crip2* mutant protein products in blue. The five base pair deletion in exon 2 (red bar), generated through CRISPR mutagenesis, results in an early stop codon in exon 3, indicated by the red asterisk. The predicted protein lacks the second half of the first LIM binding domain. (H) Schematic depicts the zebrafish *crip3* gene in black and the wild-type and predicted *crip3* mutant protein products in blue. The 11 base pair deletion in exon 2 (red bar), generated through CRISPR mutagenesis, results in an early stop codon in exon 3, indicated by the red asterisk. The predicted protein lacks the second half of the first LIM binding domain. (I,J) Representative live images of wild-type (*n*=10) (I) and *crip*$^{DM}$ (*n*=10) (J) embryos at 52 hpf. Lateral views, anterior to the left. Scale bar: 250 µm. (K,L) Quantitative PCR analyses of *crip2* (K) and *crip3* (L) expression in wild-type and *crip*$^{DM}$ embryos at 26 hpf shows significant reduction in both genes, employing unpaired, two-tailed nonparametric *t*-tests (**$P$=0.0067 in K and ****$P$<0.0001 in L). Mean and standard error of each dataset are shown.

## Crip genes are required for HSPC emergence from the dorsal aorta

Definitive hematopoiesis in zebrafish produces HSPCs that have the potential for self-renewal and the ability to differentiate into all blood cell fates (Nik et al., 2017; Orkin and Zon, 2008). To determine whether Crip genes play roles in the establishment of the definitive HSPC pool, we inspected HSPC markers, *runx1* at 26 hpf and *cmyb* (also known as *myb*) at 32 hpf, and found a statistically significant decrease in expression in *crip*$^{DM}$ compared with wild-type embryos (Fig. 2A-F). We next evaluated *cmyb* ISH in *crip2*$^{-/-}$ and *crip3*$^{-/-}$ embryos independently to examine for synergistic effects in the *crip*$^{DM}$ embryos. Our data revealed no statistically significant decrement in *cmyb* expression when comparing *crip2*$^{-/-}$ and *crip3*$^{-/-}$ with wild-type embryos (Fig. S3A-E). These findings indicate that compensation occurs between *crip2* and *crip3* genes and underscore the benefits of working with the *crip*$^{DM}$ model.

We then inspected the developing HSPCs that are derived from the aortic HE in the VDA and emerge through EHT. Employing *Tg(kdrl:mCherry);Tg(cd41:GFP)* double transgenic embryos at 40 hpf as previously described (Clements et al., 2011), our data illuminate a statistically significant decrease in the number of *kdrl*:mCherry$^+$*cd41*:GFP$^+$ cells when comparing wild-type and *crip*$^{DM}$ embryos (Fig. 2G-I). Taken together, these results implicate

a requirement for Crip genes in promoting HSPC emergence in the zebrafish embryo.

## Loss of Crip gene function impairs production of hematopoietic derivatives

We sought to explore the origins of this HSPC defect through evaluation of the primitive hematopoietic wave. Early myelopoiesis originates in the anterior lateral plate mesoderm (ALPM) and early erythropoiesis arises from the intermediate cell mass (ICM) (Davidson et al., 2003; Detrich et al., 1995); these two primary embryonic phases of blood production constitute primitive hematopoiesis (Chen and Zon, 2009). Furthermore, fate mapping studies identify a hemangioblast population in zebrafish that give rise to myeloid and endothelial progenitors (Vogeli et al., 2006; Warga et al., 2009). We examined *stem cell leukemia factor* (*tal1*) in the posterior mesoderm bilateral strips to detect these hemangioblast precursors in wild-type and *crip*$^{DM}$ embryos at 14 somites (Gering et al., 1998; Liao et al., 1998). Given normal patterning of *tal1* expression in the posterior lateral plate mesoderm (PLPM) (Fig. S4A,B), we evaluated hemangioblast derivatives such as *gata1a* and *spi1b* (also known as *pu.1*) which are expressed in erythroid and myeloid progenitors of the primitive wave of hematopoiesis, respectively (Fig. S4C-F) (Bennett et al., 2001; Davidson and Zon, 2004; Detrich et al., 1995; Lieschke

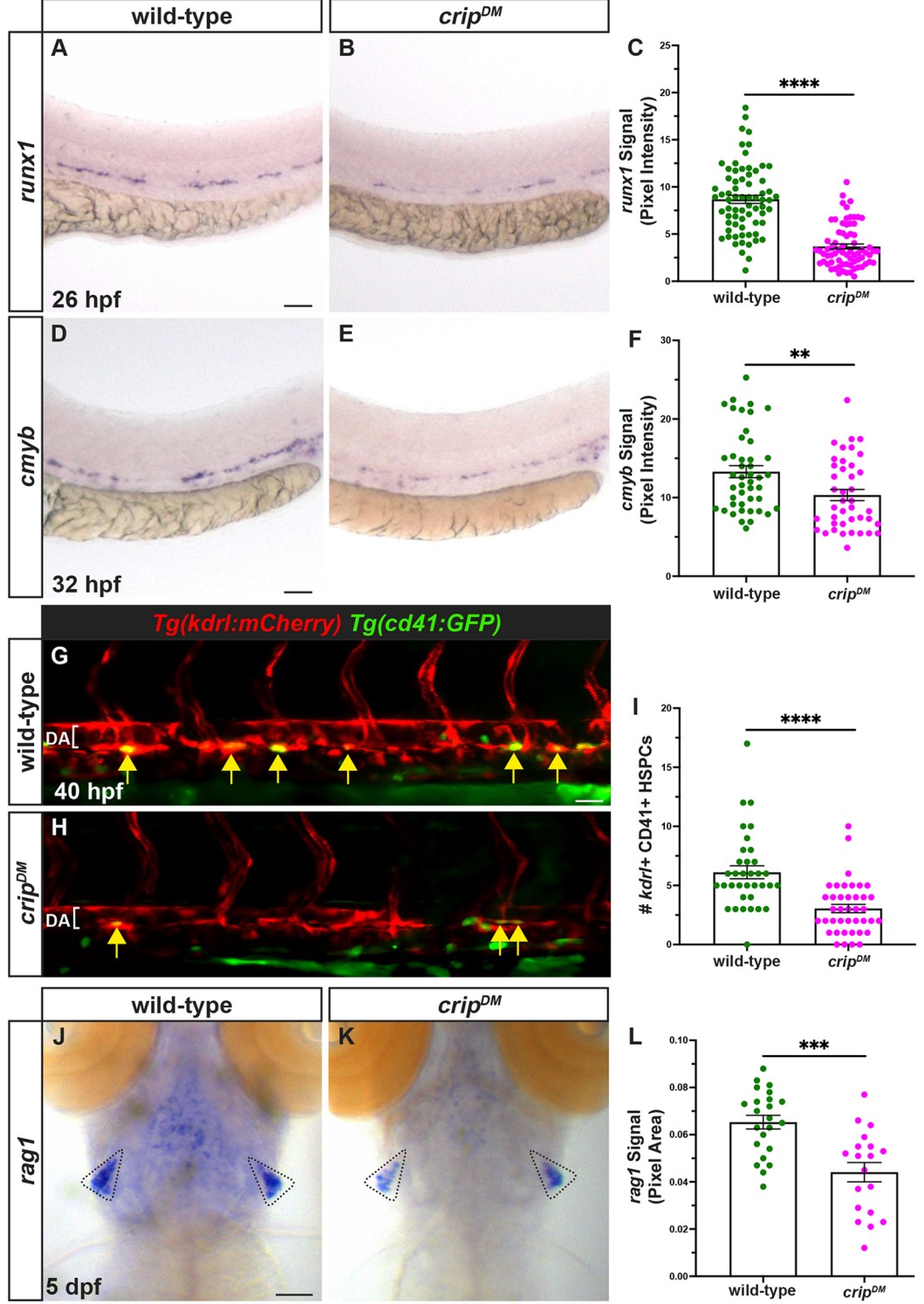

**Fig. 2. *crip2* loss-of-function results in decreased HSPCs and lineage derivatives.** (A,B) *In situ* hybridization (ISH) for *runx1* in wild-type (*n*=70) (A) and *crip^DM^* (*n*=71) (B) embryos at 26 hpf. Lateral views, anterior to the left. Scale bar: 50 µm. (C) Quantification of *runx1* signal in the dorsal aorta (DA) from A and B in wild-type and *crip^DM^* embryos using pixel intensity analysis. Unpaired two-tailed nonparametric *t*-test yields a statistically significant difference between wild-type and *crip^DM^* (****$P$<0.0001) embryos. (D,E) ISH for *cmyb* in wild-type (*n*=45) (D) and *crip^DM^* (*n*=41) (E) embryos at 32 hpf. Lateral views, anterior to the left. Scale bar: 50 µm. (F) Quantification of *cmyb* signal in the DA from D and E in wild-type and *crip^DM^* embryos using pixel intensity analysis shows a statistically significant reduced signal in *crip^DM^* embryos employing an unpaired two-tailed nonparametric *t*-test (**$P$=0.0049). (G,H) Live transgenic images of wild-type (*n*=22) (G) and *crip^DM^* (*n*=36) (H) embryos carrying *Tg(kdrl:mCherry);Tg(CD41:GFP)* at 40 hpf. Lateral views, anterior to the left. Scale bar: 25 µm. (I) Quantification of HSPCs in wild-type and *crip^DM^* embryos denoted by *kdrl*:mCherry^+^*cd41*:GFP^+^ cells in G and H. Unpaired two-tailed nonparametric *t*-test yields a statistically significant difference between wild-type and *crip^DM^* embryos (****$P$<0.0001). (J,K) ISH for *rag1* in wild-type (*n*=23) (J) and *crip^DM^* (*n*=19) (K) embryos at 5 dpf. Ventral views, anterior to the top. Dotted triangles indicate *rag1* expression. Scale bar: 50 µm. (L) Quantification of the *rag1*^+^ area in the thymus of embryos from J and K reveals decreased *rag1*^+^ area in *crip^DM^* embryos, employing an unpaired two-tailed nonparametric *t*-test (***$P$<0.001). Mean and standard error of each dataset are shown.

et al., 2002). Our investigation of *gata1a* and *spi1b* indicated no evidence of expression domain perturbations in the *crip^DM* compared to wild-type embryos. Altogether, these data document normal progression of the primitive wave of blood development in the *crip^DM* embryos.

Given that the first evidence of aberrant developmental hematopoiesis in the Crip gene loss-of-function model occurred during HSPC production (Fig. 2), we probed the ability of these progenitors to produce lineage derivatives. HSPCs travel and colonize the thymus where T lymphocytes are manufactured (Trede and Zon, 1998). Moreover, thymic seeding reflects the efficiency of definitive HSPC emergence from the dorsal aorta (Jin et al., 2007; Murayama et al., 2006). We observed a smaller field of expression of *rag1*, a thymocyte marker, in *crip^DM* compared to wild-type embryos at 5 days post fertilization (dpf) (Fig. 2J-L). Thus, our data illuminate that *crip2* and *crip3* are necessary for HSPC generation and definitive hematopoiesis with persistent requirement through T cell accumulation in the thymus.

### *crip^DM* embryos recover from deficits in HSPC production

Given the observed early insufficient definitive hematopoiesis in *crip^DM* embryos, we asked whether there were long term consequences of the depleted HSPC numbers. Thus, we tracked *cmyb* expression in wild-type compared with *crip^DM* embryos from 32 hpf and detected a recovery of the HSPC deficit in the *crip^DM* embryos by 4 dpf (Fig. 3A-H). Quantification of the *cmyb* pixel intensity at each timepoint in both genotypes corroborates normalization of expression in the *crip^DM* embryos to the wild-type levels (Fig. 3I-L). To confirm the presence and proportions of multilineage derivatives in the adult, we performed whole kidney marrow dissection with flow cytometric and fluorescence imaging quantification at 3-4 months post fertilization. These data validate no statistically significant differences in the major blood cell populations (erythroid, myeloid, lymphoid and precursors) between the wild-type and *crip^DM* zebrafish (Fig. 3M,N). Aligned with previous results emphasizing the long term viability of *runx1^−/−* embryos (Sood et al., 2010), *crip2* and *crip3* are dispensable for multilineage hematopoietic potential. Similarly, a missense mutation in *nuclear receptor subfamily 3, group C, member 1 (nr3c1)* (Ziv et al., 2013), a glucocorticoid receptor, produces *GR^s357* embryos with diminished *cmyb* expression in the CHT and T cell marker expression in the thymus (Kwan et al., 2016). Yet, *GR^s357* zebrafish survive and are morphologically indistinguishable from their wild-type siblings into adulthood (Ziv et al., 2013). Taken together, we conclude that Crip genes are vital for embryonic HSPC production while compensatory mechanisms override this deficit during expansion and maturation of adult HSPC population.

### Crip genes are required to repress endothelial cell identity

To explore how Crip genes regulate HSPC generation, we investigated the vascular endothelium given the essential role of arterial specification in HSPC formation (de Bruijn et al., 2000; Garcia-Porrero et al., 1995). We assessed *in vivo* expression patterns of pan-endothelial markers (*cdh5* and *fli1*) (Fig. 4A,B,D,E) (Lundin et al., 2020) and an arterial marker (*efnb2a*) (Fig. 4G,H) (Kobayashi et al., 2020) at 26 hpf, because the initiation of *crip2* expression occurs in the VDA at this stage. The intensity of expression of these genes was statistically significantly upregulated in *crip^DM* compared with wild-type embryos (Fig. 4C,F,I), with visible enhancement in both the dorsal aorta and the intersegmental vessels. However, the gene expression configuration of *cdh5, fli1* and *efnb2a* was appropriately patterned in *crip^DM* embryos, indicating that vasculogenesis was not

disrupted. Together, our data point to a fundamental role for Crip genes in regulating the expression levels of the vascular gene signature.

We next probed the molecular processes mediated by Crip genes in establishing the vascular endothelium using 10x Genomics scRNA-seq analysis on sorted mCherry^+ cells from *Tg(kdrl:mCherry)* and *crip^DM;Tg(kdrl:mCherry)* embryos at 30 hpf (Fig. S5A). Using the 10x Genomics Cell Ranger pipeline, we identified 7692 and 5232 cells for wild-type and *crip^DM* samples, respectively (Table S3). We annotated cell types based on differentially expressed marker genes and by using cell type markers derived from the literature (Fig. S5B,C; Table S4). From the 32 unsupervised clusters, we classified cells belonging to the nervous system (clusters 22, 24, 25, 29), epithelium (clusters 2, 21, 27, 28, 30), cardiovascular (cluster 8), endothelium (clusters 1, 3, 4, 5, 9, 11, 12, 13, 15, 19), blood (clusters 10, 16, 17, 18, 20, 26), erythroid (cluster 0) and fibroblast (clusters 6, 7, 14, 23, 31).

We then focused our interest exclusively on ECs, identifying five different populations: venous (EC venous) (clusters 1, 3, 8), arterial-venous (clusters 2, 6, 9, 15), arterial (clusters 7, 13), venous proliferating (cluster 4) and arterial-venous proliferating (cluster 14) (Fig. 5A,B; Table S5). The decrease of *crip2* gene expression was confirmed in *crip^DM* compared to wild-type ECs (Fig. S5D). We noticed that *crip^DM* ECs, as compared to wild-type, had significant upregulation of vasculature genes such as *notch1a, fli1* and *flt4*, whereas neutrophil (*lyz*) and erythroid hemoglobin (*hbbe1.1, hbbe1.3, hbbe3, hbae1.3, hbae3*) genes were significantly downregulated (Fig. 5C; Table S6). Additionally, all endothelial cell types had significantly increased average expression of these vasculature genes, computed as a higher vasculature development score (Fig. 5D; Table S7). Supporting these findings, Gene Ontology analysis showed that blood vessel development pathways, such as blood vessel remodeling, morphogenesis and differentiation, were also enriched in arterial ECs in *crip^DM* compared with wild-type embryos (Fig. 5E). Altogether, the Crip loss-of-function model skews the EC gene expression profile in favor of vasculature development, suggesting a potential disruption in the normal EC formation that is necessary for hematopoietic emergence.

### Crip-dependent inhibition of Notch is necessary to ensure HSPC generation

To dissect the mechanism underlying the upregulation of vascular signatures in the EC population in *crip^DM* embryos, we queried the molecular pathways that are misregulated in the scRNA-seq data. We directed our attention to the Notch signals, given their essential function in HSPC specification (Butko et al., 2016; Thambyrajah and Bigas, 2022). Previous studies have shown that while Notch signaling regulates HE specification and HSPC emergence, Notch is not continuously expressed (Gama-Norton et al., 2015; Li et al., 2022; Lizama et al., 2015; Zhang et al., 2017, 2015). Arterial ECs in the dorsal aorta must lose expression of arterial genes and downregulate Notch in order to gain hemogenic identity (Li et al., 2022; Zhang et al., 2017). Our findings highlighted *in vivo* upregulation of the Notch ligand, *delta-like 4 (dll4)*, and receptor, *notch1b*, in the VDA at this crucial timepoint (30 hpf) (Fig. 5F-K). These data indicate that consistent overexpression of Notch in the arterial endothelial population may result in impaired hemogenic identity establishment in the Crip loss-of-function model.

Given the robust expression of *crip2* in the dorsal aorta at the time of HE specification (Fig. 1B,C), we hypothesized that Crip genes are responsible for restricting Notch expression to repress the vascular gene network, a process necessary for EHT and HSPC generation. To confirm this theory, we assessed *Tg(kdrl:mCherry);*

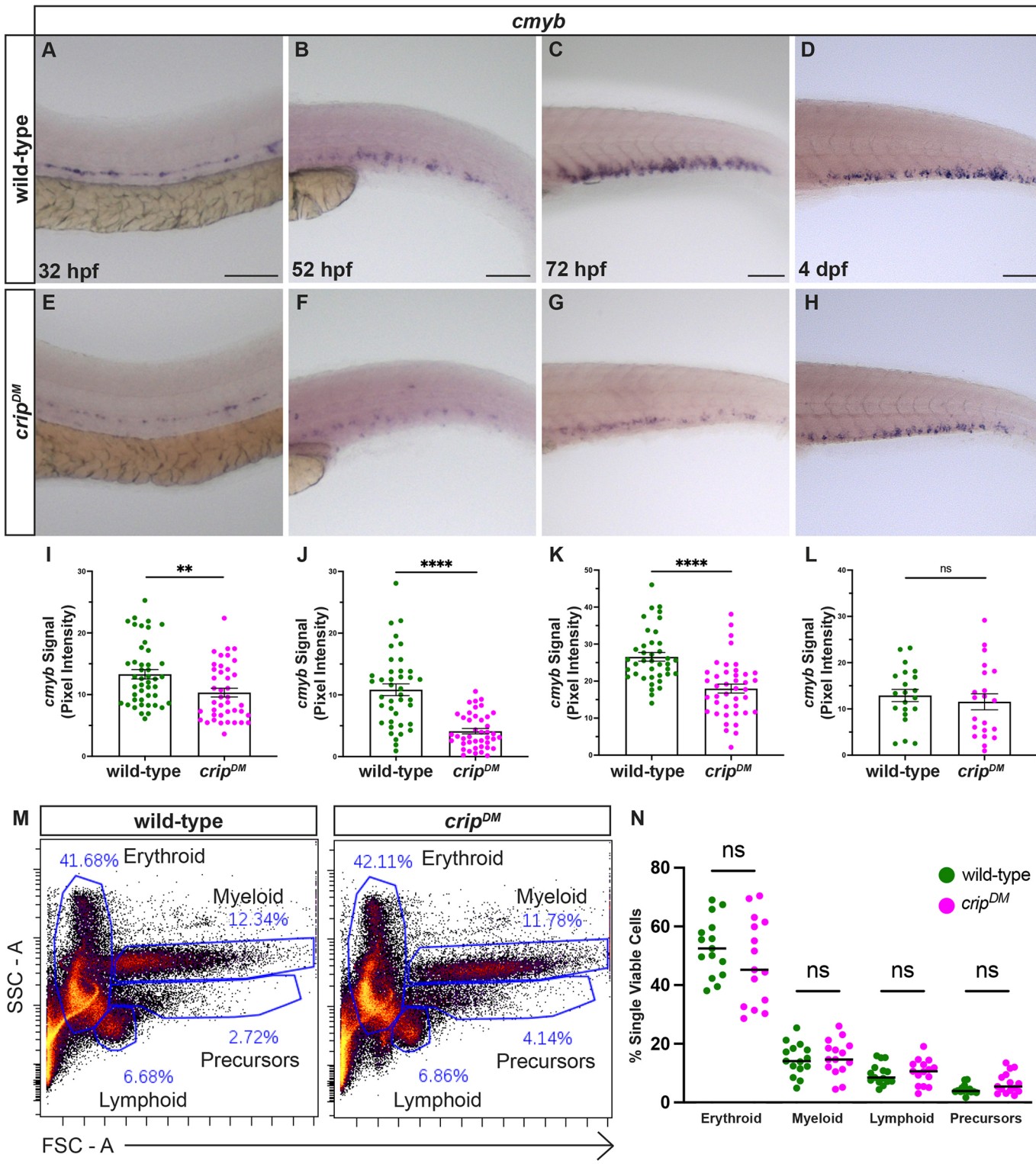

**Fig. 3. *crip^DM* embryos recover from embryonic HSPC defects.** (A-H) *In situ* hybridization for *cmyb* in wild-type (A-D) and *crip^DM* (E-H) embryos. Wild-type (*n*=45) (A) and *crip^DM* (*n*=41) (E) embryos at 32 hpf, wild-type (*n*=40) (B) and *crip^DM* (*n*=42) (F) embryos at 52 hpf, wild-type (*n*=39) (C) and *crip^DM* (*n*=41) (G) embryos at 72 hpf, and wild-type (*n*=21) (D) and *crip^DM* (*n*=21) (F) embryos at 4 dpf. Lateral views, anterior to the left. Scale bars: 100 μm. (I,J) Quantification of *cmyb* signal from A, E, B and F using pixel intensity analysis shows a reduced signal in *crip^DM* embryos employing unpaired nonparametric Mann–Whitney *U*-tests (**$P$=0.0049 in I and ****$P$<0.0001 in J). (K) Quantification of *cmyb* signal from C and G using pixel intensity analysis shows a reduced signal in *crip^DM* embryos employing an unpaired two-tailed nonparametric *t*-test (****$P$<0.001). (L) A similar comparison from D and H illustrates no statistically significant difference (ns) between wild-type and *crip^DM* embryos. Mean and standard error of each dataset are shown. (M) Representative flow cytometry analysis of whole kidney marrow (*n*=15 for wild-type and *n*=15 for *crip^DM* fish). Forward and side scatter parameters were used to define the major blood cell populations (erythroid, myeloid, lymphoid and precursors). (N) Quantification of the frequency of blood cell populations isolated from whole kidney marrow. Mean of each dataset is shown.

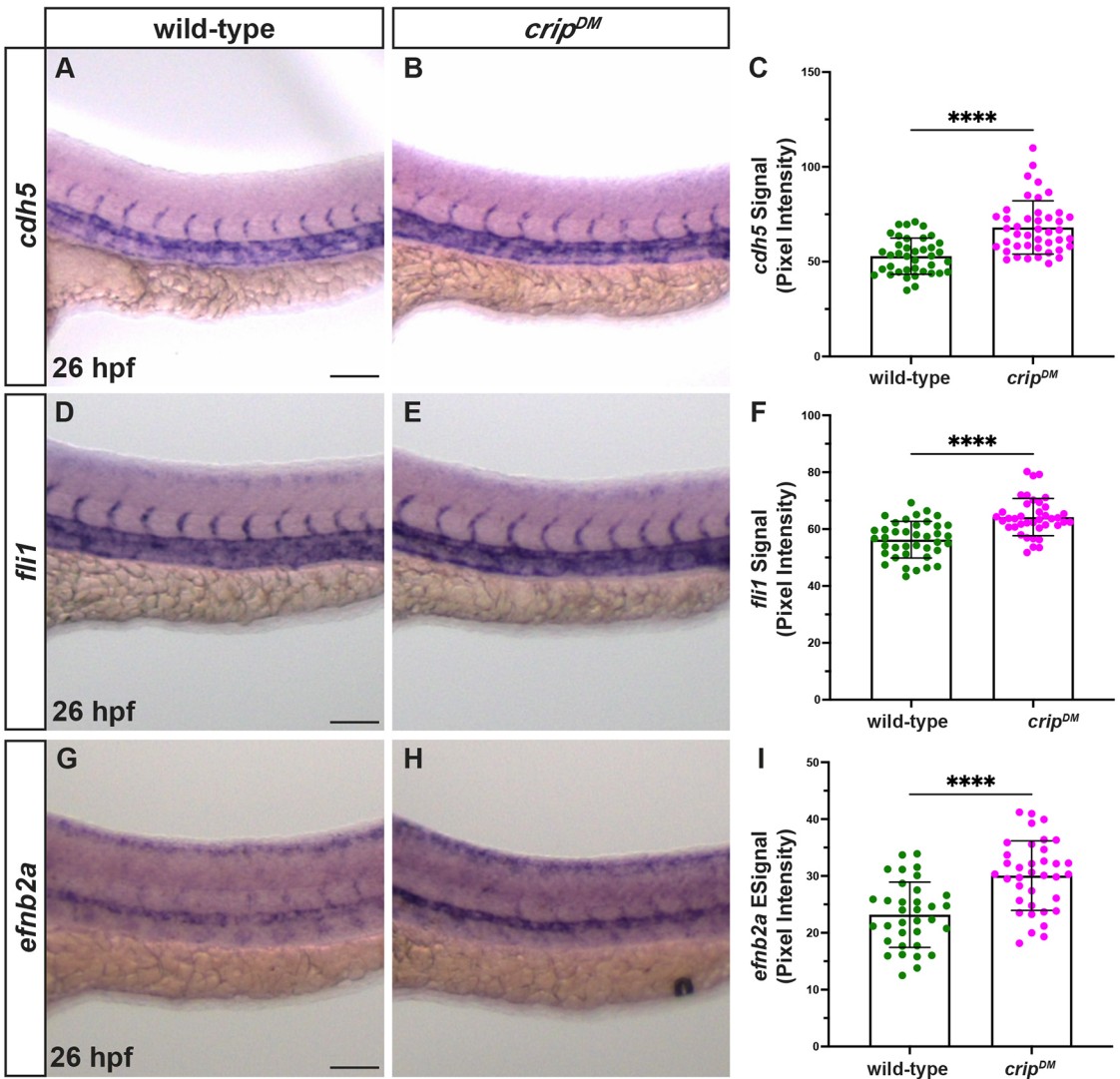

**Fig. 4. *crip^DM* embryos exhibit upregulation of the vascular gene signature.** (A-C) *In situ* hybridization (ISH) for *cdh5* in wild-type (*n*=40) (A) and *crip^DM* (*n*=41) (B) embryos at 26 hpf. Lateral views, anterior to the left. Quantification of *cdh5* signal in the dorsal aorta (DA) using pixel intensity analysis (C) shows enhanced signal in *crip^DM* embryos employing an unpaired nonparametric Mann–Whitney *U*-test (****P<0.0001). (D-F) ISH for *fli1* in wild-type (*n*=56) (D) and *crip^DM* (*n*=57) (E) embryos at 26 hpf. Lateral views, anterior to the left. Quantification of *fli1* signal in the DA using pixel intensity analysis (F). An unpaired two-tailed nonparametric *t*-test yields a statistically significant difference between wild-type and *crip^DM* (****P<0.0001) embryos. (G-I) ISH for *efnb2a* in wild-type (*n*=54) (G) and *crip^DM* (*n*=53) (H) embryos at 26 hpf. Lateral views, anterior to the left. Quantification of *efnb2a* signal in the DA using pixel intensity analysis (I). An unpaired two-tailed nonparametric *t*-test yields a statistically significant difference between wild-type and *crip^DM* (****P<0.0001) embryos. Mean and standard error of each dataset are shown. Scale bars: 100 μm.

*Tg(tp1:EGFP)* and *crip^DM;Tg(kdrl:mCherry);Tg(tp1:EGFP)* embryos treated with DMSO and found a statistically significant augmentation of *kdrl*:mCherry⁺*tp1*:EGFP⁺ cells in the ventral wall of the dorsal aorta in the loss-of-function model (Fig. 6A-C). We then exposed wild-type and *crip^DM* embryos to dibenzazepine (DBZ), a γ-secretase inhibitor that prevents processing of the Notch receptor into an active form, from 24 hpf to 32 hpf (Fig. 6D), as previously described (Li et al., 2022; Zhang et al., 2015). We observed a statistically significant rescue of the inappropriately increased number of *kdrl*:mCherry⁺*tp1*:EGFP⁺ cells in the *crip^DM* embryos following DBZ application, which corroborates effective inhibition of Notch in the HE (Fig. 6G-I). Despite absence of *kdrl*:mCherry⁺*tp1*:EGFP⁺ cellular loss in the DBZ-treated wild-type embryos (Fig. 6E,F,I), we identified a diminution in *cmyb* pixel intensity in this cohort (Fig. 6J-L). This effect on the *cmyb*-expressing HSPC population in wild-type embryos most likely represents Notch-independent,

γ-secretase-mediated cleavage of off-target proteins, as previously described (Ables et al., 2011; Yang et al., 2011). However, suppression of Notch signals by DBZ administration clearly yielded a concurrent alleviation of *cmyb*-expressing HSPC knockdown in the DBZ-treated *crip^DM* embryos when compared to the DMSO-treated *crip^DM* embryos (Fig. 6J,M,N). Taken together, these results elucidate that elevated Notch signals in the ECs of the dorsal aorta contribute to a loss of HSPC emergence in the *crip^DM* HE.

## NF-κB downstream of Crip gene function mediates Notch repression

In order to dissect how Crip proteins facilitate restriction of the Notch pathway, we invoked recent studies documenting oscillatory dynamics of NF-κB activity in the dorsal aorta that mirror transitions in Notch signals (Campbell et al., 2024; Cheng et al., 2023). Moreover, employing functional complementation in cancer cells contributing to

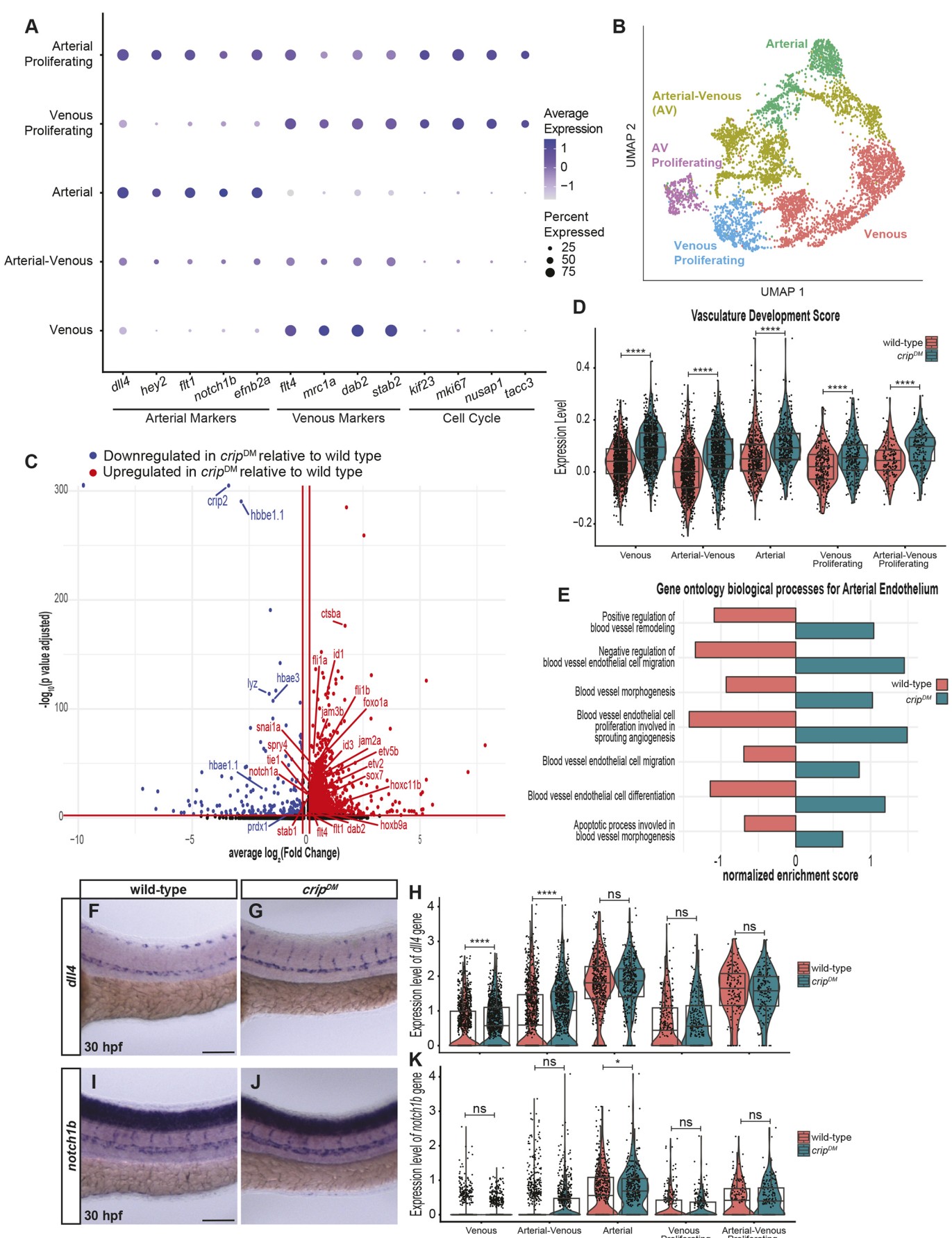

**Fig. 5.** See next page for legend.

**Fig. 5. Enriched vascular gene profile emerges in *crip^DM* embryos.**
(A) Dot plot representation for the cell type identification of endothelial populations using key arterial (*dll4, hey2, flt1, notch1b, efnb2a*), venous (*flt4, mrc1a, dab2, stab2*) and cell cycle (*kif23, mki67, nusap1, tacc3*) marker genes. Cell types include endothelial venous (EC venous), arterial-venous (AV), endothelial arterial (EC arterial), endothelial venous proliferating and endothelial arterial proliferating. Expression bar illustrates scaling performed per gene where the mean is close to 0 and the standard deviation is 1. Only genes with the highest scaled expression value will be dark purple (s.d.>1) and the lowest scaled expression will be light purple (s.d.<−1). Percent of cells expressing gene in the population is depicted by the size of the dot. (B) Cell type classifications of ECs in UMAP space as identified in A. (C) Volcano plot for upregulated (red, right) and downregulated (blue, left) genes between wild-type and *crip^DM* embryos in total EC population. Horizontal red line is at a -$\log_{10}$(*P*-value adjusted) value of 0.005 and vertical red lines are at a mean $\log_2$(fold change) value of −0.15 and 0.15. (D) Violin plots for the vasculature development gene expression score for the endothelial populations in wild-type and *crip^DM* embryos. Pairwise Wilcox test (*P*<0.0001). (E) Gene Ontology enrichment scores between wild-type and *crip^DM* samples for arterial ECs. (F-K) *In situ* hybridization for *dll4* in wild-type (*n*=41) (F) and *crip^DM* (*n*=41) (G) embryos and for *notch1b* in wild-type (*n*=38) (I) and *crip^DM* (*n*=40) (J) embryos at 30 hpf. Lateral views, anterior to the left. Scale bars: 50 μm. Violin plots for *dll4* (H) and *notch1b* (K) gene expression in scRNA-seq dataset for endothelial populations in wild-type and *crip^DM* embryos. Pairwise Wilcox test was used for these analyses (H,K) and yields statistically significant differences in noted venous (****$P$<0.0001), arterial-venous (****$P$<0.0001) and arterial (*$P$<0.05) populations. ns, not significant.

tumor development, *CRIP2* was identified as a tumor suppressor gene in nasopharyngeal carcinoma in human cell lines (Cheung et al., 2011). Data exhibiting nuclear localization of CRIP2 protein and physical interaction between *CRIP2* and NF-κB reinforce the conclusion that CRIP2 functions as a repressor of NF-κB transcriptional activity (Cheung et al., 2011). Further, studies have underscored the requirement of NF-κB transcription factor in maintenance and homeostasis of HSPCs (Espin-Palazon et al., 2014) and stress the intricate crosstalk between NF-κB and Notch networks (Oeckinghaus et al., 2011). Therefore, we evaluated the premise that Crip2 directly inhibits NF-κB to limit Notch in the dorsal aorta during HSPC emergence. Employing a previously validated NF-κB inhibitor, Caffeic acid phenethyl ester (CAPE) (Natarajan et al., 1996), we showed that CAPE application alleviates the loss of *cmyb*-expressing HSPCs in DMSO-treated *crip^DM* versus wild-type embryos, highlighting a phenotypic rescue in the Crip loss-of-function model via NF-κB inhibition (Fig. 7A-F). We subsequently tested whether repression of the NF-κB pathway functions through Notch signals given literature illuminating positive regulation of both Notch ligands and targets by NF-κB (Bash et al., 1999; Moran et al., 2007). Our data depicted a statistically significant decrement in *kdrl*:mCherry⁺*tp1:* EGFP⁺ cells in CAPE-treated *crip^DM* embryos when compared to those exposed to DMSO alone (Fig. 7G-K). Moreover, Notch signals established wild-type levels in *crip^DM* embryos following CAPE inhibition of NF-κB (Fig. 7G). This result substantiates the ability of NF-κB repression to block transcription of Notch targets. Altogether, our experiments emphasize that Crip proteins function upstream of NF-κB to suppress Notch signals at the crucial window when HSPCs arise from the dorsal aorta.

## DISCUSSION
Our findings present previously unseen insights into the pivotal role played by the Crip family members in orchestrating HSPC production during the definitive wave of embryonic hematopoiesis. Within the dorsal aorta, *crip2* emerges as a crucial mediator, influencing the expression of arterial gene signatures as ECs undergo EHT. Our

scRNA-seq data, coupled with *in vivo* results, reveal upregulated vascular signatures and heightened Notch signaling, suggesting an imbalance between arterial and hemogenic endothelial identities in the VDA as noted in the schematic model (Fig. S6). Notably, hampering Notch signaling in *crip^DM* embryos leads to a significant rescue of the HSPC population and NF-κB inhibition results in a similar restoration of these *cmyb*-expressing cells. In summary, our analyses establish *crip2* as a pivotal regulator of the arterial gene network via the Notch signaling pathway, critically influencing HSPC production within the HE.

Studies in mouse and chick models demonstrate focal downregulation of Notch activity in hematopoietic clusters with persistence of Notch signals in the neighboring aortic endothelium (Del Monte et al., 2007; Richard et al., 2013). Furthermore, diminished VE-Cadherin and Dll4 expression reinforce the conclusion that hematopoietic clusters lose arterial and endothelial identity as they undergo EHT (Richard et al., 2013). While this essential process of Notch inhibition in the HE has been observed, little is known about the upstream regulatory mechanisms. Data suggest that inhibition of Notch activity can be modulated by the strength of receptor-ligand interactions (Gama-Norton et al., 2015) and by degradation of Notch receptors via G-protein coupled receptor 183-induced proteosome pathway (Zhang et al., 2015). More recently, investigation of the DNA methylome landscape transitions exposes essential functions of DNA methyltransferase 1 (Dnmt1) in repressing Notch genes to promote the transition from EC to HSPC during EHT (Li et al., 2022). These complex and heterogenous Notch regulatory mechanisms suggest that a multi-faceted network is required to titrate pathway activity and generate HSPCs in the HE. Therefore, it is essential to dissect the range of potential inputs that fine tune this cell fate switch. Our data pinpoint Crip2 as a newly identified repressor of arterial fate in the HE that is expressed during a crucial transition of HSPC emergence. These results also delineate a previously unappreciated function of Crip2 as a vital upstream factor in curbing Notch signaling to promote the establishment of the HE at the expense of vascular fate.

Intricate cross talk between the NF-κB and Notch signaling pathways hint at the possibility that Crip2 may regulate Notch inhibition via NF-κB repression (Mishra et al., 2021; Osipo et al., 2008). Specific examples include transcriptional upregulation of *Jagged1*, a Notch ligand, and stimulation of *Hes-5* and *Deltex-1*, Notch targets, by NF-κB, both occurring in B cells (Bash et al., 1999; Moran et al., 2007). In support of this mechanism, our data encapsulate mitigation of impaired hemogenic endothelial identity following repression of NF-κB along with reestablishment of wild-type *kdrl*:mCherry⁺*cd41*:GFP⁺ cell number in *crip^DM* embryos. Given the direct repression of NF-κB by CRIP2 (Cheung et al., 2011), we hypothesize that NF-κB-responsive transcriptional activation of the Notch pathway operates downstream of Crip genes to define the HE and foster HSPC emergence.

Ultimately, translating insights and applying innovative techniques generated in animal models to differentiate human pluripotent stem cells into transplantable HSPCs will expedite therapeutic intervention in patients. Despite recent advances, our ability to facilitate maturation of HSPCs beyond embryonic stages *in vitro* is still lacking (Calvanese and Mikkola, 2023; Ding et al., 2023). Our efforts to characterize the specific, temporally-controlled requirement for Crip2, a novel regulator of HSPC emergence, in Notch repression in the HE represents a step forward in overcoming the impediments to this long-standing, unsolved problem. Although open questions are outstanding regarding the function of Crip2 in hematopoietic disorders, previous reports have shed light on therapeutic strategies delineating the importance of Crip2 recruitment by a novel long non-coding

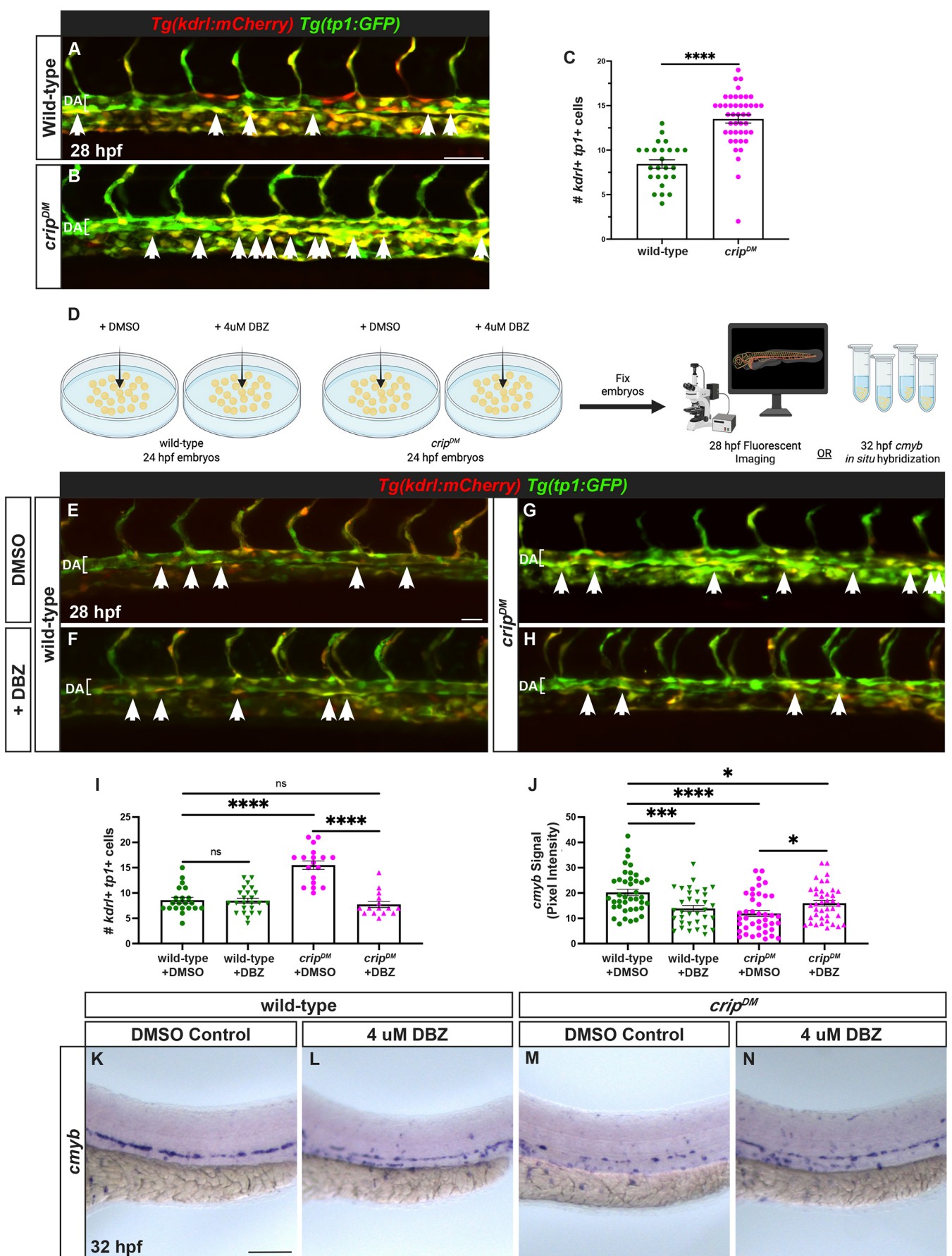

**Fig. 6.** See next page for legend.

**Fig. 6. Crip2 functions upstream of Notch signaling in the dorsal aorta to safeguard HSPC production.** (A,B) Confocal images of wild-type (*n*=25) (A) and *crip^DM* (*n*=44) (B) embryos carrying *Tg(kdrl:mCherry);Tg(tp1:EGFP)* at 28 hpf. Lateral views, anterior to the left. Arrows indicate *kdrl*: mCherry^+tp1:EGFP^+ double positive cells in the floor of the dorsal aorta. Scale bar: 50 µm. (C) Quantification of *kdrl*:mCherry^+*tp1*:EGFP^+ cells in wild-type and *crip^DM* embryos. Unpaired two-tailed nonparametric *t*-test yields a statistically significant difference between wild-type and *crip^DM* embryos (****P<0.0001). (D) Experimental design schematic depicts the addition of DMSO and DBZ to wild-type and *crip^DM* embryos at 24 hpf with fixation at 28 hpf for confocal microscopy or 32 hpf for *cmyb in situ* hybridization (ISH). (E-H) Confocal images of DMSO-treated wild-type (*n*=21) (E) and *crip^DM* (*n*=24) (G) and DBZ-treated wild-type (*n*=18) (F) and *crip^DM* (*n*=15) (H) embryos carrying *Tg(kdrl:mCherry);Tg(tp1:EGFP)* at 28 hpf reveal normalization of *kdrl*:mCherry^+*tp1*:EGFP^+ cell number in DBZ-exposed *crip^DM* embryos. Lateral views, anterior to the left. Arrows indicate *kdrl*:mCherry^+tp1:EGFP^+ double positive cells in the floor of the dorsal aorta. Scale bar: 25 µm. (I) Quantification of *kdrl*:mCherry^+*tp1*:EGFP^+ cells in the dorsal aorta (DA) from E-H shows a statistically significant reduction in the DBZ-exposed compared to the DMSO-exposed *crip^DM* embryos. Unpaired nonparametric Mann–Whitney *U*-test yields ****P<0.0001 between DMSO-exposed wild-type and *crip^DM* embryos and ****P<0.0001 between DMSO- and DBZ-exposed *crip^DM* embryos. (J-N) ISH for *cmyb* at 32 hpf illuminates reappearance of HSPCs in DBZ-exposed *crip^DM* (*n*=39) (N) compared to DMSO-exposed *crip^DM* (*n*=41) (M) embryos, delineating a trend towards HSPC numbers in DMSO-exposed wild-type (*n*=40) (K) embryos. Lateral views, anterior to the left. Scale bar: 100 µm. Quantification of *cmyb* signals in the DA from K-N using pixel intensity analysis (J) shows a statistically significant elevation in the DBZ-exposed compared to the DMSO-exposed *crip^DM* embryos. Unpaired nonparametric Mann–Whitney *U*-test yields ***P=0.0007 between DMSO- and DBZ-exposed wild-type embryos, ****P<0.0001 between DMSO-exposed wild-type and *crip^DM* embryos, *P=0.0111 between DMSO- and DBZ-exposed *crip^DM* embryos, and *P=0.0278 between DMSO-exposed wild-type and DBZ-exposed *crip^DM* embryos. Mean and standard error of each dataset are shown. ns, not significant.

RNA (lncRNA), Sarrah (SCOT1-antisense RNA regulated during aging in the heart), in galvanizing cardioprotective gene expression (Trembinski et al., 2020). Furthermore, recent work emphasizes the crucial role of *cis* inhibition of Notch pathway via Jagged1 to preserve hematopoietic stem cell fate in vertebrates (Thambyrajah et al., 2024), confirming mathematical models and validating the concept of a heterogeneous HE (Dignum et al., 2021; Nandagopal et al., 2019; Uenishi et al., 2018). One can speculate that Crip2 may participate in demarcation of the hemogenic endothelial fate through inhibition of the Notch pathway via NF-κB transcriptional repression. Harnessing insights from our data and these studies will enrich our understanding of the molecular mechanisms mediated by Crip2 and the processes by which Crip2 fine tunes Notch signals in the endothelium, ultimately supporting the discovery of tools to repopulate fully functional definitive blood precursors in diseased states.

## MATERIALS AND METHODS
### Zebrafish transgenes
We used zebrafish carrying the previously described transgenes: *Tg(-5.1myl7:nDsRed2)^f2* (RRID: ZDB-ALT-060821-6) (Mably et al., 2003), *Tg(kdrl:GFP)^la116* (RRID: ZFIN_ZDB-ALT-070529-1) (Choi et al., 2007), *Tg(kdrl:mCherry)^y206* (RRID: ZDB_ALT-111104-1) (Gore et al., 2011), *Tg(-6.0itga2b:EGFP)^la2* or *Tg(cd41:eGFP)* (RRID: ZDB_ ALT-190821-1) (Lin et al., 2005), and *Tg(tp1-MmHbb:EGFP)^um14* or *Tg(tp1:EGFP)* (RRID: ZDB_ALT-090625-1) (Parsons et al., 2009). All zebrafish experiments were performed according to protocols approved by the Institutional Animal Care and Use Committee (IACUC) at Columbia University.

### Generation of *crip2* and *crip3* mutant lines
We generated *crip2* and *crip3* alleles by targeting the first LIM binding domain of each gene. sgRNAs (*crip2*: 5′-TAATACGACTCACTATAG-GGCTGCAGTAAGACCCTGAGTTTTAGAGCTAGAAATAGCAAG-3′

and *crip3*: 5′-TAATACGACTCACTATAGGCTCCTGGAGGACATG-CGGGTTTTAGAGCTAGAAATAGCAAG-3′) were transcribed from PCR-generated templates using the MEGAscript T7 Kit (Thermo Fisher Scientific), as previously described (Gagnon et al., 2014). The NotI-digested pCS2-nCas9n plasmid (Addgene plasmid #47929) was used as a template to transcribe Cas9 mRNA with the SP6 mMESSAGE kit (Invitrogen). One-cell-stage embryos were injected with a mixture of 100 ng/µl of each gene-specific sgRNA and 300 ng/µl Cas9 mRNA. CRISPR-Cas9-injected embryos were raised to adulthood and founders (F0) were identified by PCR using the strategy outlined in the Genotyping section of these methods. Stable lines were established at the F2 generation and identified as *crip2^fcu2* (5 bp deletion in exon 2, yielding a truncated, 45 aa protein, as in Fig. 1G) and *crip3^fcu3* (11 bp deletion in exon 2, yielding a truncated, 63 aa protein, as in Fig. 1H).

### Genotyping
PCR genotyping was performed on genomic DNA extracted from embryos or individual adult tail clips. Detection of *crip2* was performed using primers 5′-CTGAGAAAGTGTCATCACTGGG-3′ and 5′-TACTCAC-CTCGGCATGGCCA-3′ to generate 102 bp wild-type and 97 bp mutant PCR fragments. Detection of *crip3* was performed using primers 5′-TCTAAAATGTGAACGATGTA-3′ and 5′-AAACAGAATGATCAGA-GAGA-3′ to generate 109 bp wild-type and 98 bp mutant PCR fragments.

### Quantitative PCR
For qPCR, total RNA was extracted from 40 pooled embryos per condition using a RNeasy Kit (Qiagen). The purified RNAs were reverse transcribed with the iScript™ cDNA synthesis kit (Bio-Rad) using 1 µg of total RNA per reaction. qPCR was performed with iQ™SYBR Green Supermix (Bio-Rad) and a CFX96 Touch™ RealTime PCR Detection System (Bio-Rad). All experiments were completed using three biological replicates. Primer sequences are noted in Table S1.

### *In situ* hybridization
We performed whole-mount ISH as previously described (Yelon et al., 1999). Removal of pigmentation on embryos older than 26 hpf is achieved through a bleaching protocol with 0.8% potassium hydroxide, 0.9% hydrogen peroxide and 0.1% Tween 20. Previously reported probes employed in this study include *runx1* (ZDB-GENE-000605-1), *cmyb* (ZDB-GENE-991110-14), *rag1* (ZDB-GENE-990415-234), *cdh5* (ZDB-GENE-040816-1), *fli1* (ZDB-GENE-980526-426), *efnb2a* (ZDB-GENE-990415-67), *dll4* (ZDB-GENE-041014-73), *notch1b* (ZDB-GENE-990415-1893), *has2* (ZDB-GENE-020828-1), *versican a* (ZDB-GENE-011023-1), *tal1* (ZDB-GENE-980526-501), *gata1a* (ZDB-GENE-980526-268) and *spi1b* (ZDB-GENE-980526-164). New probes generated for these experiments include *crip1* (ZDB-GENE-041111-1), *crip2* (ZDB-GENE-040426-2889) and *crip3* (ZDB-GENE-130530-620). Primer sequences are noted in Table S2.

### Immunofluorescence
Whole-mount double immunofluorescence was conducted using a protocol described previously (Alexander et al., 1998). Briefly, embryos were dechorionated and fixed in 1% methanol-free formaldehyde (Poly Sciences, 04018) for 1 h at room temperature, then rinsed twice in PBS and blocked with 2 mg/ml bovine serum albumin (Sigma-Aldrich, A8022), 10% goat serum (Jackson ImmunoResearch, 005000121) and 0.2% saponin (Sigma-Aldrich, S4521) in PBS for 2 h at room temperature. The embryos were incubated overnight at 4°C with primary antibodies in 0.2% saponin/PBS solution with the following dilutions: mouse anti-S46 (Developmental Studies Hybridoma Bank, 1:20), rabbit anti-dsRed (Clonetech, 632496, 1:500), mouse anti-MF20 (Developmental Studies Hybridoma Bank, 1:20) or rabbit anti-Elnb (1:200, Song et al., 2019). The next day, embryos were washed twice in 0.2% saponin/PBS solution and incubated for 2 h at room temperature with secondary antibodies in 0.2% saponin/PBS with the following dilutions: anti-mouse IgG2b Alexa Fluor 568 (Invitrogen, A21144, 1:200), anti-rabbit IgG Alexa Fluor 488 (Invitrogen, A11034, 1:200), anti-rabbit IgG Alexa Fluor 568 (Invitrogen, A11011, 1:375) or anti-mouse IgG1 Alexa Fluor 488 (Invitrogen, A21121, 1:200). DAPI (1:1000) was added as needed during the

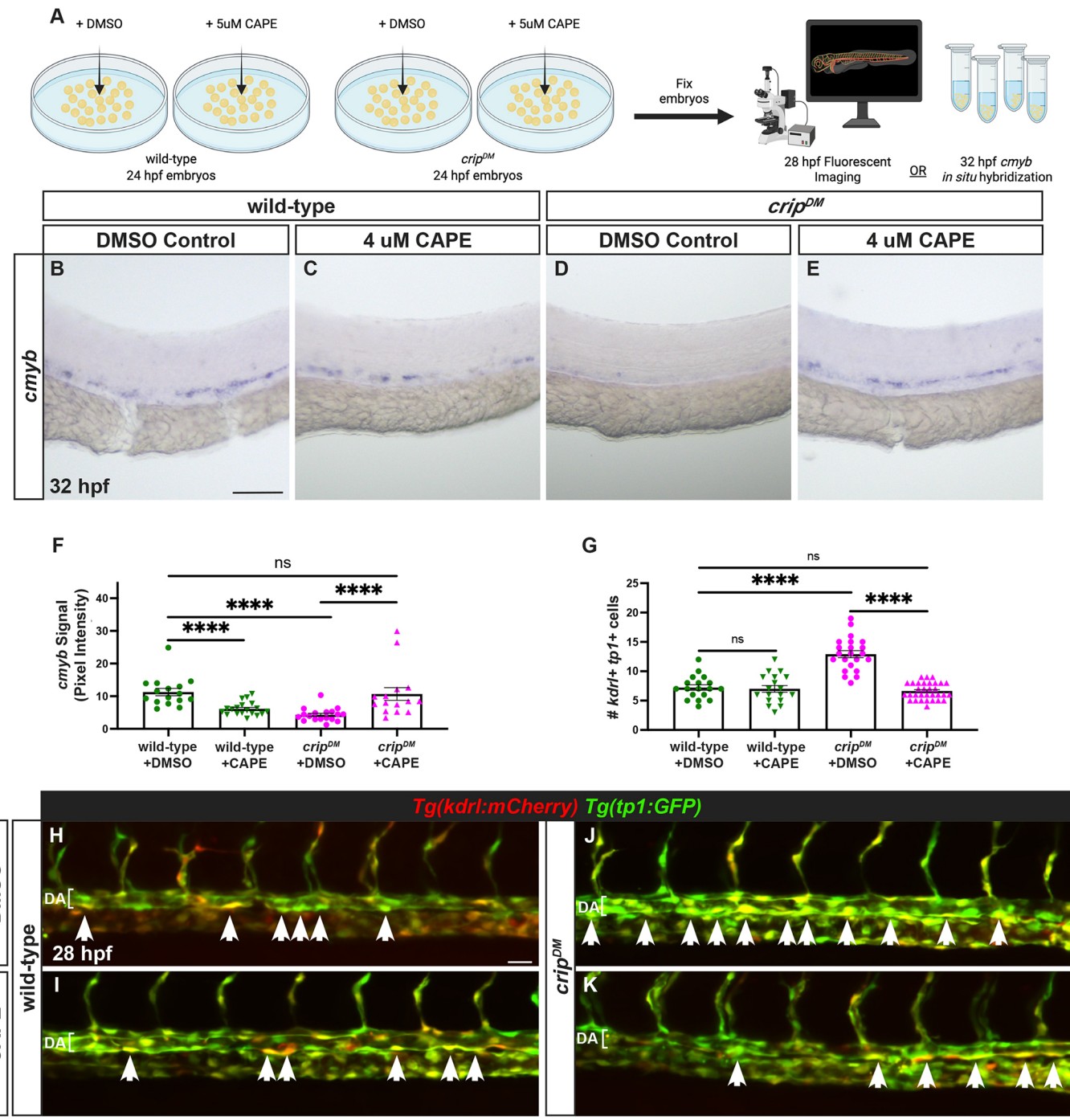

**Fig. 7. NF-κB regulates Notch targets downstream of Crip genes.** (A) Experimental design schematic depicts the addition of DMSO and CAPE to wild-type and *crip^DM* embryos at 24 hpf with fixation at 28 hpf for confocal microscopy or at 32 hpf for *cmyb in situ* hybridization (ISH). (B-E) ISH for *cmyb* at 32 hpf illuminates reemergence of HSPCs in CAPE-exposed (*n*=15) (E) compared to DMSO-exposed (*n*=18) (D) *crip^DM* embryos. Images from DMSO- (*n*=16) (B) and CAPE-exposed (*n*=19) (C) wild-type embryos are also depicted. Lateral views, anterior to the left. Scale bar: 100 μm. (F) Quantification of *cmyb* signals in the dorsal aorta (DA) from B-E using pixel intensity analysis shows a statistically significant boost in the CAPE-exposed compared to the DMSO-exposed *crip^DM* embryos. Unpaired nonparametric Mann–Whitney *U*-test yields ****$P<0.0001$ between DMSO- and CAPE-exposed wild-type embryos, between DMSO-exposed wild-type and *crip^DM* embryos, and between DMSO- and CAPE-exposed *crip^DM* embryos. (G-K) Confocal images of DMSO-treated wild-type (*n*=18) (H) and *crip^DM* (*n*=22) (J) and CAPE-treated wild-type (*n*=18) (I) and *crip^DM* (*n*=32) (K) embryos carrying *Tg(kdrl:mCherry);Tg(tp1:EGFP)* at 28 hpf underscore a rescue of *kdrl*:mCherry⁺*tp1*:EGFP⁺ cell number in CAPE-exposed *crip^DM* embryos. Lateral views, anterior to the left. Arrows indicate *kdrl*:mCherry⁺tp1:EGFP⁺ double positive cells in the floor of the dorsal aorta. Scale bar: 25 μm. Quantification of *kdrl*:mCherry⁺*tp1*:EGFP⁺ cells in H-K (G) shows a statistically significant decrease in the CAPE-exposed compared to the DMSO-exposed *crip^DM* embryos. Unpaired nonparametric Mann–Whitney *U*-test yields ****$P<0.0001$ between DMSO-exposed wild-type and *crip^DM* embryos and between DMSO- and CAPE-exposed *crip^DM* embryos. Mean and standard error of each dataset are shown. ns, not significant.

secondary antibody incubation. Finally, embryos were rinsed three times in 0.2% saponin/PBS before imaging.

## Imaging

Images for live embryos, immunofluorescence and ISH experiments were captured using a Zeiss M2Bio microscope and a Zeiss AxioCam digital camera, before processing with Zeiss AxioVision and Adobe Creative Suite software. MF20/Elnb images were acquired on a Zeiss 710 laser scanning confocal microscope with a Zeiss AXIO Observer Z1 inverted microscope stand. Confocal imaging of transgenic embryos was performed on an A1 confocal scanner mounted on a TiE Eclipse stand (Nikon Instruments). Following data acquisition, all z-stacks were analyzed using ImageJ (version 1.53k, National Institutes of Health).

## Cardiomyocyte counting

To quantify cardiomyocytes as previously described (Targoff et al., 2008), immunofluorescence was performed to detect DsRed in cardiomyocyte nuclei of Tg(-5.1myl7:nDsRed2)-carrying embryos and S46 in the atrium. Embryos were gently flattened between a slide and a coverslip in preparation for imaging. Individual cardiomyocytes were counted using ImageJ software on reconstructed z-stacks. Student's t-test (homoscedastic, two-tailed distribution) determined statistical significance between the means of cell number datasets.

## Adult kidney marrow isolation

For adult kidney marrow analysis, kidneys from zebrafish aged 3-4 months were dissected, resuspended in 500 ml of FACS buffer supplemented with 1 mg/ml DAPI, and filtered through a 40 mm cell strainer (Falcon). Forward and side scatter parameters were used to resolve the major blood cell lineages of erythroid, myeloid, lymphoid and precursors, as previously described (Traver et al., 2003). All samples were analyzed at the Flow Cytometry Core Facility at the Albert Einstein College of Medicine using an LSR II flow cytometer (BD Biosciences) and data were processed using FlowJo Software (versions 10.6-10.8).

## Statistical analysis and quantification

All quantifications were performed on anonymized data. Results represent at least two independent experiments (technical replicates) in which multiple embryos from multiple independent matings are analyzed (biological replicates). Statistical analysis was performed using GraphPad Prism Version 9.5.0. Normal distribution was assessed using a Shapiro-Wilk test. A two-tailed Student's t-test was used to determine significance for all continuous variables that were normally distributed. The Mann–Whitney U-test was used to determine significance for all continuous variables that were not normally distributed. Fisher's exact test was used when data involved categorical variables. Statistical values are displayed as mean ±standard error of the mean (s.e.m.). The following nomenclature was employed to present results: ns, not significant; $*P<0.05$; $**P<0.01$; $***P<0.001$; $****P<0.0001$. For quantification of the ISH pixel intensity signals, previously reported protocols were executed (Dobrzycki et al., 2018).

## Single cell RNA-sequencing

Cellular dissociation was performed on 30 Tg(kdrl:mCherry)$^{y206}$ and 30 crip2$^{-/-}$;crip3$^{-/-}$;Tg(kdrl:mCherry)$^{y206}$ embryos at 30 hpf with 5 mg/ml liberase (Roche) solution in PBS at 37°C, pipetting every 5 min until adequately homogenized in solution. Fetal bovine serum (FBS) (5% of volume) was then added to each sample to arrest the dissociation followed by centrifugation (500 g for 5 min) at 4°C. The excess supernatant was removed and the pellet was resuspended in PBS/1% FBS and filtered through a 40 μm cell strainer. Finally, DAPI (1:1000) was added and cells were sorted using a Special Order Research Product FACSAria™ Cell Sorter (BD Biosciences) under gentle conditions with the 130 μm nozzle at 12 PSI. NERL Diluent 2 (Thermo Fisher Scientific, DIL5522) solution was used for sheath fluid. FACSAria was calibrated per the standard protocol in the CSCI Flow Cytometry Core Facility using Cytometer Setup and Tracking Beads (BD Biosciences, 655051) and SPHERO Rainbow Calibration Particles (8 Peaks, 3.0 μm) (Spherotech, RCP-30-5A), followed

by optimization of the drop charge delay using BD FACS Accudrop Beads (BD Biosciences, 345249).

Cells were selected to be sorted based on the following scheme: first, forward and side light scatter (488 nm) were used to identify cells and establish an initial gate to remove debris. Following this selection, single cells were identified by plotting forward light scatter pulse area against forward scatter pulse height. Signals from DAPI (measured with 405 nm excitation at 100 mW and a 450/50 bandpass filter), a cell impermeant nuclear dye, were used to select live, nucleated cells based on low DAPI signal. Finally, mCherry signal was collected alongside these other parameters using 561 nm excitation at 100 mW and a 610/20 bandpass filter.

Single cells were captured to prepare the library for the sequencing using the Chromium Single Cell 3′ Reagent Kit (10x Genomics, version 3.1) and the 10x Genomics platform according to the manufacturer's instructions. Following assessment of quality and size with the Bioanalyzer, the cDNA libraries were sequenced on a NovaSeq 6000 at the JP Sulzberger Columbia Genome Center at Columbia University Medical Center.

## Mapping and clustering of single cell mRNA data

Fastq files of wild-type and crip2$^{-/-}$;crip3$^{-/-}$ (hereafter referred to as crip$^{DM}$) samples were aligned to a custom zebrafish reference genome (Danio rerio GRCz11/danRer11, version 105) which includes the mCherry and GFP sequences built using the 10x Genomics Cell Ranger software (version 3.1.0) (Zheng et al., 2017) with default parameters (Table S3). The output of Cell Ranger was analyzed using Seurat (version 4.1.0) (Butler et al., 2018; Hao et al., 2021; Satija et al., 2015; Stuart et al., 2019). The feature-barcode matrices of the two samples were read into an R environment using the Read10X() function. A Seurat object was created for each sample using the CreateSeuratObject() function and merged. The percentage of mitochondrial gene counts were computed using the PercentageFeatureSet() function. A subset of high-quality cells was selected for analysis, defined as cells expressing greater than 200 genes (nFeature_RNA>200), overall mitochondrial gene count lower than 10% (percent.mt<10), and greater than 50 unique molecular identifiers detected (nCount_RNA>50). This filtering strategy yielded a total of 12,924 cells across both samples.

Wild-type and crip$^{DM}$ sample integration was performed as described in the Seurat vignette (Stuart et al., 2019). Briefly, the standard workflow was completed on the merged Seurat object [NormalizeData(), FindVariableFeatures(), ScaleData(), RunPCA(), etc.] and layers were merged with JoinLayers(). Clustering was completed with the FindNeighbors() and FindClusters() functions using 43 principal components (PCs) and a resolution parameter of 1.1, which resulted in 32 clusters. These results were visualized in a two-dimensional UMAP representation produced using the DimPlot() function.

## Cell type identification in single cell mRNA data

We used cell type markers from the literature to determine the identities of cell clusters in wild-type and crip$^{DM}$ samples. The following cell types were identified: nervous system (clusters 22, 24, 25, 29), epithelium (clusters 2, 21, 27, 28, 30), cardiovascular (cluster 8), endothelium (clusters 1, 3, 4, 5, 9, 11, 12, 13, 15, 19), blood (clusters 10, 16, 17, 18, 20, 26), erythroid (cluster 0) and fibroblast (clusters 6, 7, 14, 23, 31) (Fig. S5B; Table S4). As expected, each cell type cluster expressed known gene markers: neuronal (neurod1, pou3f1, dpysl5a, ank2b); epithelial (epcam, krt92, cyt1l); fibroblast (pdgfra, prrx1a, prrx1b, twist1a); heart (tbx20, gata4, gata5, fn1a); endothelial (kdrl, etv2, cldn5b, fli1); hemoglobin (hbbe1.1, hbbe1.2, hbbe1.3, hbbe2, hbbe3); blood (coro1a, ptprc); neutrophil (mpx, lyz); and macrophage (irf8, mfap4.1, mpeg1.1) (Fig. S5C).

We subset the data to select only endothelial populations for downstream analysis. We reprocessed this new subset data using the following workflow: ScaleData(); RunPCA(); RunUMAP() with reduction 'pca' and 45 dimensions; and FindClusters with a resolution of 0.7, which yielded 18 clusters. We used the following gene markers to guide our cell type identification: arterial (dll4, hey2, flt1, notch1b, efnb2a); venous (flt4, mrc1a, dab2, stab2); and cell cycle (kif23, mki67, nusap1, tacc3). The following endothelial cell types were identified: venous (EC venous) (clusters 1, 3, 8); arterial-venous (clusters 2, 6, 9, 15); arterial (clusters 7, 13); venous proliferating (cluster 4); and arterial-venous proliferating (cluster 14) (Table S5).

## Differential gene expression analysis and Gene Ontology

FindMarkers() function from Seurat with Wilcoxon rank sum test was used to perform differential gene expression analysis between wild-type and $crip^{DM}$ cells for endothelial cell types. For the Wilcoxon test, an adjusted $P$-value <0.005 and a log fold change threshold minimum of 0.1 were used to select for differentially expressed genes (DEGs) (Table S6). The gene ontology annotations for *Danio rerio* were taken from Biomart and the average expression of genes for each GO term was computed, as previously described (Durinck et al., 2009; Hunter et al., 2021).

## Zebrafish chemical treatments

We applied a previously published protocol describing the use of the γ-secretase inhibitor DBZ (Barske et al., 2016) to block processing of the Notch receptor into its active intracellular form (Ichida et al., 2014). Specifically, DBZ (Tocris, 4489) was dissolved in dimethyl sulfoxide (DMSO) (Thermo Fisher Scientific) creating a 100 mM stock which was diluted to a final concentration of 4 µM in embryo medium. No more than 40 embryos were incubated in a single dish starting at 24 hpf. DBZ was thoroughly rinsed with embryo medium and embryos were fixed at 32 hpf. Finally, paired control wild-type and $crip^{DM}$ embryos were exposed to the same concentration of DMSO for the equivalent developmental windows for each respective experiment.

CAPE, an inhibitor of NF-κB (Natarajan et al., 1996), was used to validate that NF-κB acts downstream of Crip genes, employing a previously published protocol (Campbell et al., 2024). We dissolved 25 mg of CAPE (Sigma-Aldrich, 211200) in 4.4 ml of DMSO to create a 20 mM stock solution. At 24 hpf, 20 to 30 embryos were incubated with 5 mM of CAPE stock diluted in embryo medium. The same concentration of DMSO was applied to paired control wild-type and $crip^{DM}$ embryos. The treatment for all conditions was arrested at 28 hpf by transferring the embryos to fresh medium and thoroughly rinsing twice.

## Acknowledgements
We thank Dr Teresa Bowman for sharing multiple tools (*gata2b*, *flt4* and *dll4* probes) and techniques and for her critical reading of the manuscript. We are grateful to Dr Wilson Clements for supplying reagents (*rag1*, *cdh5* and *efnb2a* probes). Dr David Traver was also instrumental in providing additional probes (*cmyb*, *spi1b* and *fli1*). We appreciate the constructive feedback for this manuscript from former members of the Targoff Laboratory, Dr Di Yao, Elizabeth Myrus and Micah Woodard, and also the expert zebrafish care by Joshua Barber. We are thankful to Michael Kissner of the Columbia Stem Cell Initiative Flow Cytometry Core, Dr Peter Sims of the JP Sulzberger Columbia Genome Center, and Drs John Murray and Munemasa Mori of the Columbia Center for Human Development Microscopy Core. We are also grateful to Drs Theresa Swayne and Istvan Boldogh in the Confocal and Specialized Microscopy Shared Resource of the Herbert Irving Comprehensive Cancer Center at Columbia University Medical Center. These imaging studies used resources that are funded in part through National Institutes of Health (NIH)/National Cancer Institute (NCI) Cancer Center Support Grant P30CA013696.

## Competing interests
The authors declare no competing or financial interests.

## Author contributions
Conceptualization: A.G.A., C.d.S.-T.; Data curation: K.L.T., A.G.A., R.N., C.F., K.S.P., C.d.S.-T., C.V., U.R., R.S., T.B.; Formal analysis: K.L.T., A.G.A., B.U., K.S.P., U.R., H.K.E., R.S., T.B.; Funding acquisition: K.L.T., A.G.A.; Investigation: K.L.T., A.G.A., R.N., T.B.; Methodology: K.L.T., A.G.A., B.U., T.B.; Supervision: K.L.T., M.G.K.; Writing – original draft: K.L.T., A.G.A., B.U.; Writing – review & editing: K.L.T., A.G.A., B.U., K.S.P., C.d.S.-T., R.S., T.B.

## Funding
This project was supported by grants to K.L.T. from the National Institutes of Health (R01 HL13143801A1), to A.G.A. from the National Institutes of Health (T32 HL120826 and F31 HL164101), to C.F. from the National Institutes of Health (F31 DE030385) and a New York State Stem Cell Science training grant, to T.B. from the National Institutes of Health (R01 DK121738 and R01 DK131445) and the Edward P. Evans Foundation, and to B.U. from the National Institutes of Health (T32 GM007288 and F31 HL152562) and the American Society of Hematology. Open Access funding provided by Columbia University. Deposited in PMC for immediate release.

## Data and resource availability
The RNA-sequencing data reported in this paper have been deposited in GEO under accession number GSE300243. All relevant data and details of resources can be found within the article and its supplementary information or may be requested from the corresponding author.

## Peer review history
The peer review history is available online at https://journals.biologists.com/dev/lookup/doi/10.1242/dev.204359.reviewer-comments.pdf

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
