## [Peer Review File · Development (Cambridge, England)]

Crip2 preserves hematopoietic stem and progenitor cell production through inhibition of Notch signals

Angelika G. Aleman, Bianca Ulloa, Rigolin Nayak, Caitlin Ford, Kathryn S. Potts, Carmen de Sena-Tomás, Camila Vicioso, Uday Rangaswamy, Harold K. Elias, Michael G. Kharas, Remo Sanges, Teresa Bowman and Kimara L. Targoff
DOI: 10.1242/dev.204359

Editor: Steve Wilson

Review timeline

Original submission:	6 September 2024
Editorial decision:	14 November 2024
First revision received:	26 September 2025
Editorial decision:	6 November 2025
Second revision received:	15 January 2026
Accepted:	19 January 2026

Original submission

First decision letter

MS ID#: dev.204359

MS Title: Crip2 preserves hematopoietic stem and progenitor cell production through inhibition of Notch signals

Authors: Angelika G. Aleman; Bianca Ulloa; Caitlin Ford; Kathryn S. Potts; Carmen de Sena-Tomás; Camila Vicioso; Uday Rangaswamy; Remo Sanges; Teresa Bowman; Kimara L. Targoff

Article Type: Research Article

Dear Kimara,

I have now received all the referees' reports on the above manuscript, and have reached a decision. The referees' comments are appended below, or you can access them online: please go to:

As you will see, the referees differ in their opinion on the suitability of the study for publication in Development. Reviewer 1 is the most critical but reviewer 3 raises a similar criticism that you need to perform additional studies to clarify the mechanistic basis of the phenotypes you describe. Reviewer 1's third comment is very open-ended and has more suggestions than you could reasonably address but taken together with the comments from Reviewer 3, there are good suggestions for how to extend the study. If you are able to address these and the other concerns and revise the manuscript along the lines suggested, I will be happy receive a revised version of the manuscript. Please also note that Development will normally permit only one round of major revision. If it would be helpful, you are welcome to contact us to discuss your revision in greater detail. Please send us a point-by-point response indicating your plans for addressing the referees' comments, and we will look over this and provide further guidance.

Please attend to all of the reviewers' comments and ensure that you clearly highlight all changes made in the revised manuscript. Please avoid using 'Tracked changes' in Word files as these are lost in PDF conversion. I should be grateful if you would also provide a point-by-point response detailing

how you have dealt with the points raised by the reviewers in the 'Response to Reviewers' box. If you do not agree with any of their criticisms or suggestions please explain clearly why this is so.

Reviewer 1

SUMMARY OF THE ADVANCE MADE IN THIS PAPER AND ITS POTENTIAL SIGNIFICANCE TO THE FIELD

In their manuscript, Aleman and colleagues show a non-described role for *crip2* in the emergence of definitive hematopoiesis in the zebrafish embryo. By examining a double mutant for *crip2* and *crip3*, they show that EHT events are reduced, concomitant to an increase of notch signaling in the aortic niche. While the description of the phenotypes is quite documented, one has no idea of the precise mechanism(s) involved in this *crip2*-dependent biological process.

SUGGESTIONS TO AUTHORS

Major points:

1- The authors claim that *crip3* compensated for the loss of *crip2*, without showing it, therefore rationalizing their need to study the double *crip2/3* mutant. It seems important to show that *crip3* is expressed abnormally in *crip2* mutants. The authors should therefore conduct *crip3* in situ hybridization in *crip2*-het incross and/or in *crip2*-ko incross. Additionally, they could also use a *crip3*-MO in the *crip2*-ko background. That, in addition to their expression patterns in wild-type embryos, would convince that *crip2* only is required for HSPC production from the aorta, as claimed in the last sentence of the results chapter entitled "*crip2* is required for HSPC emergence from the dorsal aorta".

2- according to their expression pattern, *crip2* is expressed in all arteries, all along the floor and roof of the dorsal aorta, but as well as in ISVs. Why should there be only a phenotype at the level of the floor of the aorta where EHT occurs?

3- The authors show by transcriptome analysis that genes related to the notch pathway are upregulated in *crip2* mutant embryos. Hence, they conclude that *crip2* downregulates notch signaling to promote EHT. There is absolutely no mechanism described, but rather a long dissertation in the discussion, with hypotheses. For the quality of this paper, it is important that the last figure represents the *crip2*-dependent mechanism, which implies that the authors need to explore further the relationships between *crip2*, Notch, and maybe NFkB and/or VEGF signaling, as *crip2* has been shown to influence/modulate all these pathways, which are also important for EHT. Indeed, it is not known if the effect of *crip2* on notch is direct or indirect, i.e. the consequence of any other effect on another pathway. I would therefore suggest to increment the tool box, with transgenic lines measuring notch activity, NFkB activity, etc... and work with different small compounds, inhibitors, full length mRNA, morpholinos, etc, in order to dissect the precise mechanisms underlying these observations, and possibly give an answer to major point 2.

Minor points:

1- First sentence of the introduction, it should be noted that HSPCs serve as the source of all "blood" cell types, rather than "all cell types", which can be misleading. Even this statement is wrong it has been demonstrated that definitive HSCs do not give rise to a plethora of tissue-resident hematopoietic cell types.

2- First paragraph of the introduction: it looks like the authors are not aware of the latest published work from Jaffredo and colleagues which shows nice HSC production from iPSCs.

3- page 24: *rag1* is not a T cell marker, as mature T cells do not express *rag1*, but rather a thymocyte marker.

4- Figure 3MN, the authors have inverted the lymphoid and precursor gates (see Traver, Nat Immunol 2003).

Reviewer 2

SUMMARY OF THE ADVANCE MADE IN THIS PAPER AND ITS POTENTIAL SIGNIFICANCE TO THE FIELD

SUGGESTIONS TO AUTHORS

The authors reported that a double mutant for *crip2/3* in zebrafish exhibited impaired definitive hematopoiesis and mechanistically they showed that *crip2/3* repressed notch signaling to ensure the EHT during HSPC emergence.

Overall, the manuscript is well written and most of data are very clean and reliable. I have a few of concerns as follows:

- 1, Regarding the underlying mechanism, at which level the Notch signaling is affected remains elusive. Previous reports have shown that Notch signaling can be regulated at multiple levels including transcriptional and posttranslational levels, what about *crip2/3*?
- 2, The recovery of HSPC phenotypes is interesting but needs a bit of discussion for clarification. Where do these HSPCs and derivatives come from after 4dpf in zebrafish?

Minor issues:

- 1, In the title, it is *crip2* but in the main work, it should be *crip2/3*.

Reviewer 3

SUMMARY OF THE ADVANCE MADE IN THIS PAPER AND ITS POTENTIAL SIGNIFICANCE TO THE FIELD

In the manuscript "Crip2 preserves hematopoietic stem and progenitor cell production through inhibition of Notch Signals", the authors demonstrate a requirement for Crip2 expression during development of the hemogenic endothelium and specification of hematopoietic stem and progenitor cells (HSPCs). The manuscript is well-written and presents elegant data characterizing the double Crip2 and Crip3 mutant and clearly demonstrating a requirement for Crip2 in HSPC specification. The requirement of Crip2 is transient, as HSPC specification is recovered at 4 days post fertilization. The mutant is unique in that the phenotype is very specific, with no major morphological changes observed in the double mutant embryos. Further characterization and evidence of Notch regulation by Crip2 is needed. If the following issues could be addressed, the manuscript may be of interest to the readers of Development.

SUGGESTIONS TO AUTHORS

1. The connection between Crip2 expression and regulation of Notch signaling does not have sufficient supporting data, especially given the significance placed on this connection in the manuscript and title. Notch signals are higher in the absence of Crip2 expression, but the manuscript lacks a direct connection. This is partially addressed in the discussion, identifying previously published data on a connection between Notch and NF- κ B through Crip2, but no mechanism is given nor proposed. If there is a connection between NF- κ B, what is the effect of NF- κ B repression and Notch expression in the mutants and rescue of HSPCs? Can the increase in notch expression be shown with in situ hybridization specifically in the dorsal aorta? Does treatment change notch inhibition in the dorsal aorta?
2. What is the effect of DBZ on WT embryos? A clear rescue is shown, but without the control treated comparison, it is difficult to conclude that the rescue is specific to the mutant phenotype.
3. Are there any gene expression differences between CripDM mutant and WT HSPCs after recovery to suggest why specification is no longer depended on Crip expression?

Minor issue

1. Images in Figure S1 are not particularly clear. Although the expression differences can be observed, the resolution is fuzzy, making them more difficult to interpret. A cartoon of heart locations above each image may also be helpful here to orient a reader not proficient with

zebrafish heart morphology. Zoomed out images could also be helpful. This is also true for S2-H-K. Images in 2J, K could be improved with circles or marks indicating the thymic regions.

First revision

Author response to reviewers' comments

Response to Reviewers - Aleman et al. (dev.204359)

We are grateful for the astute and constructive comments provided by the reviewers and appreciate the feedback that “*the mutant is unique in that the phenotype is very specific*” and that “*the manuscript is well written and most of data are very clean and reliable*”. Furthermore, the reviewers comment that our story “*presents elegant data characterizing the double Crip2 and Crip3 mutant and clearly demonstrates a requirement for Crip2 in HSPC specification*”.

The feedback outlined by each reviewer prompted significant revisions to our manuscript and we believe these modifications have substantially improved our study. We comprehensively and systematically respond to each conceptual concern and experimental question below.

Reviewer #1

Major points

1) *The authors claim that crip3 compensated for the loss of crip2, without showing it, therefore rationalizing their need to study the double crip2/3 mutant. It seems important to show that crip3 is expressed abnormally in crip2 mutants. The authors should therefore conduct crip3 in situ hybridization in crip2-het incross and/or in crip2-ko incross. Additionally, they could also use a crip3-MO in the crip2-ko background. That, in addition to their expression patterns in wild-type embryos, would convince that crip2 only is required for HSPC production from the aorta, as claimed in the last sentence of the results chapter entitled “crip2 is required for HSPC emergence from the dorsal aorta”.*

We agree that additional data is necessary to support our claim that studying the *crip2^{-/-};crip3^{-/-}* (*crip^{DM}*) embryos enhances our analysis. Thus, we performed *crip2^{+/-}* intercross to generate both wild-type and *crip2^{-/-}* embryos in the same clutch, per Reviewer #1's suggestion, and implemented *in situ* hybridization (ISH) with the *crip3* probe. Surprisingly, our results illuminate no significant elevation in *crip3* expression. Given that standard techniques for mRNA transcript detection are not robust enough to reveal subtle, yet critical modulation of gene expression, we conclude that there still may be functional compensation between *crip2* and *crip3*. Findings presented in the original Fig. S3 demonstrate statistically significant exacerbation of decreased *cmyp⁺* HSPCs in *crip^{DM}* when compared with *crip2^{-/-}* and *crip3^{-/-}* embryos. These data underscore the benefit of executing our experiments with *crip^{DM}* to obviate any redundancy between the two *Crip* genes.

2) *According to their expression pattern, crip2 is expressed in all arteries, all along the floor and roof of the dorsal aorta, as well as in ISVs. Why should there be only a phenotype at the level of the floor of the aorta where EHT occurs?*

We thank Reviewer #1 for this astute comment and have since evaluated myriad features of the *crip^{DM}* phenotype to explain consequences of the loss of *crip2* expression throughout the dorsal

aorta and the intersegmental vessels (ISVs). In summary, we conclude that the statistically significant upregulation of the vascular gene signature (*cdh5*, *fli1*, and *efnb2a*) throughout the dorsal aorta and the ISVs (Fig. 4) illustrates increased arterial fate secondary to diminished *crip2* expression in this entire domain. While the HSPC emergence defect highlights the failure of Crip2 to repress the hemogenic endothelium at the expense of vascular fate, long term outcomes of this transient boost in arterial identity do not result in vasculogenesis patterning abnormalities.

3) The authors show by transcriptome analysis that genes related to the notch pathway are upregulated in crip2 mutant embryos. Hence, they conclude that crip2 downregulates notch signaling to promote EHT. There is absolutely no mechanism described, but rather a long dissertation in the discussion, with hypotheses. For the quality of this paper, it is important that the last figure represents the crip2-dependent mechanism, which implies that the authors need to explore further the relationships between crip2, Notch, and maybe NFkB and/or VEGF signaling, as crip2 has been shown to influence/modulate all these pathways, which are also important for EHT. Indeed, it is not known if the effect of crip2 on notch is direct or indirect, i.e. the consequence of any other effect on another pathway. I would therefore suggest to increment the toolbox, with transgenic lines measuring notch activity, NFkB activity, etc... and work with different small compounds, inhibitors, full length mRNA, morpholinos, etc, in order to dissect the precise mechanisms underlying these observations, and possibly give an answer to major point 2.

We are grateful for the suggestion to probe deeper into the *crip2*-dependent mechanisms. To this end, we implemented several additional experiments employing an augmented toolbox. We utilized a Notch-responsive transgene, *Tg(tp1:MmHbb:EGFP)^{um14}*, to compare Notch signals in the dorsal aorta in wild-type versus *crip2^{DM}* embryos at 28 hpf. We detect statistically significantly upregulated numbers of *kdrl:mCherry⁺tp1:EGFP⁺* cells in the ventral wall of the dorsal aorta in our loss-of-function model (new Fig. 6A-C). Moreover, following DBZ treatment, this enhanced Notch signal was diminished which is consistent with our findings that Notch inhibition is sufficient to rescue the HSPC emergence defect (new Fig. 6E-N). Next, we examined potential interactions with the NF-κB pathway through inhibition via CAPE administration. Our new Fig. 7 underscores the requirement of NF-κB functioning downstream of Crip2 to restrict Notch signals (new Fig. 7G-K). Moreover, delivery of CAPE successfully rescues the HSPC deficiency (new Fig. 7B-F). We merge these additional experiments and their mechanistic insights into the body of the manuscript on pages 29-32 to support our conclusion that Crip2 physically interacts with NF-κB to repress transcription of Notch target genes during HSPC emergence.

Minor points:

1) First sentence of the introduction, it should be noted that HSPCs serve as the source of all "blood" cell types, rather than "all cell types", which can be misleading. Even this statement is wrong it has been demonstrated that definitive HSCs do not give rise to a plethora of tissue-resident hematopoietic cell types.

We concur that the first sentence of the introduction was misleading and we modified the text to acknowledge the complexity of the cell types derived from the HSPC population.

2) First paragraph of the introduction: it looks like the authors are not aware of the latest published work from Jaffredo and colleagues which shows nice HSC production from iPSCs.

We appreciate the important of including recent studies from the Jaffredo Laboratory and have incorporated these findings into our introduction.

3) Page 24: rag1 is not a T cell marker, as mature T cells do not express rag1, but rather a thymocyte marker.

We updated this sentence to correct this error in our description of *rag1*.

4) Figure 3MN, the authors have inverted the lymphoid and precursor gates (see Traver, Nat Immunol 2003).

We thank Reviewer #1 for pointing out this oversight and have corrected the lymphoid and precursor gates per Traver et al., *Nat Immunol* 2003.

Reviewer #2

Major points

1) Regarding the underlying mechanism, at which level the Notch signaling is affected remains elusive. Previous reports have shown that Notch signaling can be regulated at multiple levels including transcriptional and posttranslational levels, what about *crip2/3*?

In order to probe the mechanisms by which Notch signaling is regulated by *Crip* genes, we crossed *Tg(tp1:MmHbb:EGFP)^{um14}* into our *crip^{DM};Tg(kdrl:mCherry)^{v206}* line to assess for transcriptional activation of Notch signaling. Specifically, this transgenic regulatory sequence directs EGFP expression upon binding of Notch intra-cellular domain (NICD) and its cofactor RBP-Jk to the *tp1* element. Our data highlight a statistically significant increase in the number of *kdrl:mCherry⁺tp1:EGFP⁺* cells when comparing *crip^{DM}* to wild-type embryos (new Fig. 6A-C). Incorporating these data, we generated a new Fig. 6 supporting our conclusion that *Crip* genes regulate Notch signaling at the transcriptional level.

2) The recovery of HSPC phenotypes is interesting but needs a bit of discussion for clarification. Where do these HSPCs and derivatives come from after 4 dpf in zebrafish?

Despite our eagerness to identify the mechanisms by which HSPCs and their derivatives recover after 4 dpf in *crip^{DM}* embryos, we believe that addressing these questions is beyond the scope of our initial publication. On page 26, we discuss similar phenomena observed in *runx1^{-/-}* (Sood et al., 2010) and *GR^{s357}* embryos (Kwan et al., 2016) where early deficits in HSPC production resolve, permitting long term survival of mutant larvae. Another example is described in Karp et al., *bioRxiv* 2024 whereby mutagenesis of *khdrbs1a/b*, an RNA binding protein, leads to a statistically significantly decreased HSPC population at 36 hpf. Yet, *cmyb⁺* cells normalize by 3 dpf, ultimately illustrating that this EHT defect does not lead to long term consequences in HSPC numbers of the CHT. These data are aligned with our results from *crip^{DM}* compared to wild-type embryos at 4 dpf as depicted in our original Fig. 4. Taken together, these examples from the field accentuate the plastic nature of HSPC production from the EHT and the resilient recovery observed in myriad genetic environments.

Minor points

1) In the title, it is *crip2* but in the main work, it should be *crip2/3*.

We agree that the main text of the manuscript should present the findings in the context of the *crip2;crip3* loss-of-function model. Therefore, we edited the document and added references to *crip2* and *crip3* and/or *Crip* genes.

Reviewer #3

Major points

1) The connection between *Crip2* expression and regulation of Notch signaling does not have sufficient supporting data, especially given the significance placed on this connection in the manuscript and title. Notch signals are higher in the absence of *Crip2* expression, but the manuscript lacks a direct connection. This is partially addressed in the discussion, identifying previously published data on a connection between Notch and NF- κ B through *Crip2*, but no mechanism is given nor proposed. If there is a connection between NF- κ B, what is the effect of NF- κ B repression and Notch expression in the mutants and rescue of HSPCs? Can the increase in notch expression be shown with *in situ* hybridization specifically in the dorsal aorta? Does treatment change notch inhibition in the dorsal aorta?

We thank Reviewer #3 for these probing questions and tackled these inquiries by performing new experiments, as mentioned in our response to Reviewer #1 (point 3). Specifically, we employed a Notch-responsive transgene, *Tg(tp1:MmHbb:EGFP)^{um14}*, to compare Notch signals in the dorsal aorta in wild-type versus *crip^{DM}* embryos at 28 hpf. We detect a statistically significantly upregulated number of *kdrl:mCherry⁺tp1:EGFP⁺* cells in the ventral wall of the dorsal aorta in our loss-of-function model (new Fig. 6A-C). These novel findings validate our *in situ* hybridization data

illuminating enhanced *notch1b* and *dll4* expression in *crip^{DM}* compared to wild-type embryos (Fig. 5F,G,I,J). Next, we show that the number of *kdrl:mCherry⁺tp1:EGFP⁺* cells are depleted following DBZ application in *crip^{DM}* embryos (new Fig. 6E-I). Finally, we examined potential interactions with the NF- κ B pathway through inhibition employing CAPE administration (new Fig. 7A). Our recently completed studies reveal a similar rescue of the *cmyb*-expressing HSPCs in CAPE- versus DMSO-treated *crip^{DM}* embryos (new Fig. 7B-F). Moreover, Notch signals establish wild-type levels in *crip^{DM}* embryos following CAPE inhibition of NF- κ B (Fig. 7G-K), substantiating the ability of NF- κ B repression to block transcription of Notch targets.

2) *What is the effect of DBZ on WT embryos? A clear rescue is shown, but without the control treated comparison, it is difficult to conclude that the rescue is specific to the mutant phenotype.*

We recognize that inclusion of the DBZ-treated wild-type embryos is essential and appreciate the prompt to integrate these data. Our new Fig. 6 includes the image and the quantitative data for these samples. Given the importance of Notch signaling on HSPC emergence and potential effect of γ -secretase inhibitors on other signaling pathways (Golde et al., *Biomembranes* 2013), we note a decrease in *cmyb* pixel intensity in wild-type embryos with DBZ. However, conversely, a boost in *cmyb* expression occurs when comparing *crip^{DM}* embryos treated with DBZ versus DMSO. Take together, these data reinforce our conclusion that inhibition of Notch signaling between 24 hpf and 32 hpf is sufficient to recuperate the diminished HSPC emergence in the absence of *Crip* genes.

3) *Are there any gene expression differences between Crip^{DM} mutant and WT HSPCs after recovery to suggest why specification is no longer depended on Crip expression?*

In order to address this question from Reviewer #3, we re-analyzed our scRNA-seq dataset (30 hpf). Specifically, we evaluated the EC subcluster for differentially expressed genes and confirmed downregulation of *crip2* in the *crip^{DM}* embryos (depicted below). However, upon further sub-clustering to isolate the rare HSPC population, our data illuminate no significant differences in *crip2* expression between wild-type and *crip^{DM}* embryos (depicted below). Together, this analysis reinforces our model that *crip2* deficiency functions in the hemogenic endothelium. Yet, following HSPC emergence, this population is no longer exposed to decreased EC *crip2* expression in the loss-of-function model and the HSPCs home and expand appropriately in the CHT.

Minor points

1) *Images in Figure S1 are not particularly clear. Although the expression differences can be observed, the resolution is fuzzy, making them more difficult to interpret. A cartoon of heart locations above each image may also be helpful here to orient a reader not proficient with zebrafish heart morphology. Zoomed out images could also be helpful. This is also true for S2-H-K. Images in 2J, K could be improved with circles or marks indicating the thymic regions.*

We value these constructive suggestions to optimize our cardiac data and to improve the accessibility of our images to all readers. Specifically, we performed additional experiments to

acquire *crip3* ISH panels in both wild-type and *crip^{DM}* embryos at 52 hpf with enhanced clarity (see new Fig. S1E,H). For the other images in the figure, we decreased the zoom to ameliorate the perspective. We also updated Fig. S2 to provide similar magnification adjustments. Finally, cartoon images are added in the new Figs. S1 and S2 to direct the readers' attention to the particular components of the cardiac anatomy highlighted in each context. In response to Reviewers #3's suggestion, we now include dotted triangles to underscore the thymic regions in the new Fig. 2J,K.

Second decision letter

MS ID#: dev.204359R1

MS Title: Crip2 preserves hematopoietic stem and progenitor cell production through inhibition of Notch signals

Authors: Angelika G. Aleman; Bianca Ulloa; Rigolin Nayak; Caitlin Ford; Kathryn S. Potts; Carmen de Sena-Tomás; Camila Vicioso; Uday Rangaswamy; Harold K. Elias; Michael G. Kharas; Remo Sanges; Teresa Bowman; Kimara L. Targoff
Article Type: Research Article

Dear Kimara,

I have now received all the referees reports on the above manuscript, and there are just a couple of points to consider/address before proceeding to publication. One of these is a suggestion to add a summary figure - I will leave this to you to decide. The referees' comments are appended below.

Reviewer 1

The authors have implemented a lot of new results in this revised version following all reviewers' comments.

The paper now shows a clear mechanism involving *crip2* (and maybe *crip3*) in the down-regulation of notch signaling, through negatively downregulating NFκB signaling.

For future manuscripts/revisions, it would be helpful that the authors highlight the changes made from the original submitted manuscript.

Also it was not really clear how they answered to the first comment about *crip2* and 3 redundancy/compensation. There was no legend attached to the picture in the rebuttal letter. Was it the heart?. It could have been better to show the aorta region, which was more relevant to the subject of the paper.

Finally, the manuscript could have benefitted from a conclusive figure with a suggested mechanism, showing the successive actions of Notch/*crip2*-NFκB/notch in the EHT process. This would help the reader, but the manuscript is still acceptable for publication in Development.

Reviewer 2

SUGGESTIONS TO AUTHORS

I have no further questions.

Reviewer 3

SUMMARY OF THE ADVANCE MADE IN THIS PAPER AND ITS POTENTIAL SIGNIFICANCE TO THE FIELD

In the manuscript "Crip2 preserves hematopoietic stem and progenitor cell production through inhibition of Notch Signals", the authors demonstrate a requirement for Crip2 expression during

development of the hemogenic endothelium and specification of hematopoietic stem and progenitor cells (HSPCs). The manuscript is well-written and presents elegant data characterizing the double *Crip2* and *Crip3* mutant and clearly demonstrating a requirement for *Crip2* in HSPC specification. The requirement of *Crip2* is transient, as HSPC specification is recovered at 4 days post fertilization. The mutant is unique in that the phenotype is very specific, with no major morphological changes observed in the double mutant embryos. The addition of the Notch response transgene data addresses the majority of concerns with this manuscript and provide strong, clear evidence of the connection between *Crip2*, Notch, and NF- κ B. Only minor issues remain.

SUGGESTIONS TO AUTHORS

1. The violin plots in figure 5H and K appear to be missing some elements of analysis.

Second revision

Author response to reviewers' comments

Response to Reviewers - Aleman et al. (dev.204359)

We appreciate the meticulous and constructive feedback offered by the Reviewers in response to our manuscript revision and are grateful for the comments describing the new version as demonstrating “*a clear mechanism*” for *Crip2* in the endothelial-to- hematopoietic transition. Furthermore, they highlight the considerable effort put forth to attend to the feedback, ultimately providing strong “*evidence of the connection between Crip2, Notch, and NF- κ B*”.

In this newly edited version, we incorporate the few additional recommendations by the Reviewers.

- 1) We include below a legend for the new data included in the Response to Reviewers to elucidate the details of the images outlining the expression of *crip3*. However, we are unable to depict the dorsal aorta in this context because *crip3* transcripts are not visualized in this domain.

crip3 is normally expressed in *crip2*^{-/-} embryos

(A,B) *In situ* hybridization for *crip3* in the hearts of wild-type (n = 17) (A) and *crip2*^{-/-} (n = 14) (B) embryos at 52 hpf. Ventral views, anterior to the top. Scale bar, 20 μ m.

- 2) We generated a schematic of the mechanistic model that we are proposing. This new figure is incorporated in Figure S6.
- 3) We present a complete analysis of the violin plots in Figure 5H,K as requested by Reviewer #3.

Third decision letter

MS ID#: dev.204359R2

MS Title: Crip2 preserves hematopoietic stem and progenitor cell production through inhibition of Notch signals

Authors: Angelika G. Aleman; Bianca Ulloa; Rigolin Nayak; Caitlin Ford; Kathryn S. Potts; Carmen de Sena-Tomás; Camila Vicioso; Uday Rangaswamy; Harold K. Elias; Michael G. Kharas; Remo Sanges; Teresa Bowman; Kimara L. Targoff

Article Type: Research Article

Dear Kimara,

I am happy to tell you that your manuscript has been accepted for publication in Development, pending our standard publication integrity checks.